# Calibration of cell-intrinsic interleukin-2 response thresholds guides design of a regulatory T cell biased agonist

Caleb R Glassman[1,2,3], Leon Su[1,3], Sonia S Majri-Morrison[1,3], Hauke Winkelmann[4], Fei Mo[5], Peng Li[5], Magdiel Pérez-Cruz[6], Peggy P Ho[7], Ievgen Koliesnik[8], Nadine Nagy[8], Tereza Hnizdilova[9], Lora K Picton[1,3], Marek Kovar[9], Paul Bollyky[8], Lawrence Steinman[7,10], Everett Meyer[6], Jacob Piehler[4], Warren J Leonard[5], K Christopher Garcia[1,3,11]*

[1]Department of Molecular and Cellular Physiology, Stanford University School of Medicine, Stanford, United States; [2]Immunology Graduate Program, Stanford University School of Medicine, Stanford, United States; [3]Department of Structural Biology, Stanford University School of Medicine, Stanford, United States; [4]Department of Biology, University of Osnabrück, Osnabrück, Germany; [5]Laboratory of Molecular Immunology and Immunology Center, National Heart, Lung, and Blood Institute, NIH, Bethesda, United States; [6]Division of Blood and Marrow Transplantation, Department of Medicine, Stanford University School of Medicine, Stanford, United States; [7]Department of Neurology and Neurological Sciences, Stanford University, Stanford, United States; [8]Division of Infectious Diseases and Geographic Medicine, Department of Medicine, Stanford University, Stanford, United States; [9]Laboratory of Tumor Immunology, Institute of Microbiology of Czech Academy of Sciences, Prague, Czech Republic; [10]Department of Pediatrics, Stanford University, Stanford, United States; [11]Howard Hughes Medical Institute, Stanford University School of Medicine, Stanford, United States

*For correspondence:
kcgarcia@stanford.edu

**Abstract** Interleukin-2 is a pleiotropic cytokine that mediates both pro- and anti-inflammatory functions. Immune cells naturally differ in their sensitivity to IL-2 due to cell type and activation state-dependent expression of receptors and signaling pathway components. To probe differences in IL-2 signaling across cell types, we used structure-based design to create and profile a series of IL-2 variants with the capacity to titrate maximum signal strength in fine increments. One of these partial agonists, IL-2-REH, specifically expanded Foxp3+ regulatory T cells with reduced activity on CD8+ T cells due to cell type-intrinsic differences in IL-2 signaling. IL-2-REH elicited cell type-dependent differences in gene expression and provided mixed therapeutic results: showing benefit in the in vivo mouse dextran sulfate sodium (DSS) model of colitis, but no therapeutic efficacy in a transfer colitis model. Our findings show that cytokine partial agonists can be used to calibrate intrinsic differences in response thresholds across responding cell types to narrow pleiotropic actions, which may be generalizable to other cytokine and growth factor systems.

## Introduction

Cytokines are secreted factors that regulate diverse aspects of cell physiology through assembly of cell surface receptors and induction of the JAK-STAT signaling cascade. Cytokines are highly pleio-tropic due to broad expression of receptor components on diverse cell types. In vivo, pleiotropy is managed by local cytokine secretion and differential expression of receptor subunits and signaling

pathway components depending on cell type and activation state. The actions of a single cytokine on multiple cell types has limited both basic understanding of cell type-dependent cytokine activity in vivo and therapeutic interventions that rely on the systemic administration of cytokine. Thus, new approaches are needed to limit cytokine pleiotropy in order to understand basic aspects of cytokine biology across cell types and develop improved therapeutic modulators of cytokine signaling.

Interleukin-2 (IL-2) is a classical pleiotropic cytokine that signals through a heterodimeric receptor formed by IL-2 receptor beta (IL-2Rβ, CD122) and common gamma chain (γ$_c$, CD132) expressed on T cells, natural killer (NK) cells, and B cells (*Hatakeyama et al., 1989*; *Takeshita et al., 1992*). Cells can further enhance their sensitivity to IL-2 by expression of the non-signaling subunit, IL-2 receptor alpha (IL-2Rα, CD25) which sensitizes cells to IL-2 signaling (*Cosman et al., 1984*; *Leonard et al., 1984*; *Nikaido et al., 1984*). Although IL-2 was originally described as a T cell stimulatory factor (*Morgan et al., 1976*), mice deficient in IL-2 or the IL-2 receptor were found to develop severe auto-immunity (*Sadlack et al., 1995*; *Sadlack et al., 1993*; *Suzuki et al., 1995*; *Willerford et al., 1995*). This seeming paradox was resolved by the finding that IL-2 signaling is critical for maintenance of regulatory T cells (Tregs) which play an important role in suppressing immune activation (*Fontenot et al., 2005*). Indeed, transfer of Tregs was shown to inhibit autoimmune syndrome in IL-2Rβ-deficient mice (*Malek et al., 2002*). Thus, IL-2 acts as a central regulator of immunity by supporting both pro- and anti-inflammatory responses.

While many immune cells express components of the IL-2 receptor, cell types differ in their requirements for and sensitivity to IL-2 signaling. NK cells express high levels of IL-2Rβ and respond to both IL-2 and the related cytokine, IL-15 which shares signaling receptors IL-2Rβ/γ$_c$ with IL-2 (*Carson et al., 1994*; *Giri et al., 1994*). Naïve T cells express low levels of IL-2 receptor components and thus are relatively insensitive to IL-2 stimulation until after upregulation of IL-2 receptor components following activation (*Au-Yeung et al., 2017*; *Leonard et al., 1985*). Unlike naïve T cells, Tregs constitutively express IL-2Rα (*Sakaguchi et al., 1995*) and are further sensitized to IL-2 by miR-155-dependent repression of suppressor of cytokine signaling 1 (SOCS1) (*Lu et al., 2015*; *Lu et al., 2009*). These differences in cell type sensitivity create threshold effects where low doses of IL-2 activate Tregs without inducing signaling on naïve T cells and NK cells while high doses of IL-2 activate both Tregs as well as naïve T cells and NK cells (*Grinberg-Bleyer et al., 2010*; *Matsuoka et al., 2013*; *Yu et al., 2015*); however, exploiting this phenomenon clinically has been difficult due to a narrow therapeutic window between pro- and anti-inflammatory IL-2 signaling (*Pham et al., 2015*).

Previous attempts to overcome IL-2 pleiotropy and target signaling to distinct cell populations have focused on differential expression of IL-2 receptor components. Antibodies against IL-2 that favor binding to IL-2Rα expressing cells (JES6-1) can potentiate Tregs by sterically blocking interactions between IL-2Rβ and IL-2 (*Boyman et al., 2006*; *Spangler et al., 2015*). Similarly, IL-2 muteins with reduced affinity for IL-2Rβ have been used to increase dependence on IL-2Rα and enhance Treg selectivity (*Khoryati et al., 2020*; *Peterson et al., 2018*; *Shanafelt et al., 2000*). While using ligands to target differences in receptor expression provides some amount of cell type selectivity, cells employ diverse mechanisms to regulate IL-2 sensitivity.

To probe intrinsic differences in IL-2 signaling across cell types, we took an unbiased and structure-based approach by generating a series of IL-2 muteins with reduced affinity for γ$_c$, which we surmised would modulate the stability of the heterodimeric signaling complex and serve to 'titrate' signal strength. We then systematically characterized their cell type specificity in vivo. Functionally, these IL-2 variants exhibit partial agonism, eliciting a submaximal response at saturating concentrations of cytokine. We identified an IL-2 mutein, IL-2-REH (L18R̲, Q22E̲, Q126H̲), that induces proliferation of Foxp3+Tregs with reduced activity on NK cells and CD8+ T cells. This mutein showed increased dependence on IL-2Rα expression and decreased capacity to overcome negative regulation by SOCS1, thereby mirroring two key features of IL-2 sensitivity in Tregs, constitutive expression of IL-2Rα and miR-155-mediated suppression of SOCS1. Expanded Tregs retain suppressive capacity in vitro and improved recovery in the dextran sulfate sodium (DSS)-induced colitis model of inflammatory bowel disease. These results demonstrate that cell type-intrinsic differences in IL-2 signaling can be exploited by partial agonists to generate new cytokines with reduced pleiotropy and improved therapeutic properties.

## Results

### IL-2 receptor partial agonists control STAT5 signaling by modulating receptor dimerization

IL-2 signaling is initiated when IL-2 first binds to IL-2Rβ, following recruitment of $\gamma_c$, which results in trans-phosphorylation of JAK1/JAK3, and activation of STAT5 and other pathway components. To systematically probe the relationship between receptor dimerization and downstream signaling, we targeted residues in IL-2 that make contact with $\gamma_c$ in order to modulate the efficiency of receptor heterodimer formation (*Wang et al., 2005*; *Figure 1A*). This mutational strategy has the advantage of capping maximal signaling (Emax) by weakening the recruitment of $\gamma_c$, the final step of IL-2 receptor assembly. We selected mutations on IL-2 that form contacts with $\gamma_c$ and installed these mutations in a variant of IL-2 with high affinity for IL-2Rβ, super-2 (H9) (*Levin et al., 2012*), which effectively 'clamps' IL-2Rβ and reduces dependence on IL-2Rα such that the Emax for the mutants are determined by the efficiency of $\gamma_c$ recruitment into the complex. For this and subsequent experiments, IL-2 variants were expressed with an N-terminal mouse serum albumin (MSA) tag to aid expression and extend half-life in vivo. A panel of H9 variants with L18R, Q22E, and variable amino acid substitutions at Q126 were tested for STAT5 signaling in YT cells, an IL-2 responsive human NK cell line, by phosphoflow cytometry. This mutational strategy yielded a panel of IL-2 partial agonists characterized by reduced maximal STAT5 phosphorylation at saturating doses of cytokine (*Figure 1B*). The diversity of response profiles from these IL-2 muteins indicates that changes to a single amino acid side chain can modulate the full range of induction of intracellular signaling, from 100% $E_{max}$ to essentially no activity.

For further characterization, we selected a smaller panel of IL-2 muteins that represented a broad range of pSTAT5 signaling intensities: H9-REH, H9-REK, and the IL-2 antagonist H9-RETR which has been previously characterized (*Mitra et al., 2015*). Based on the mutational strategy used, we expect IL-2 partial agonists to reduce receptor dimerization due to weakened interactions with $\gamma_c$. To monitor receptor dimerization, we used dual-color single-molecule particle co-tracking of IL-2Rβ and $\gamma_c$ in the plasma membrane of living cells by total internal reflection fluorescence (TIRF) microscopy. HeLa cells were transfected with IL-2Rβ and $\gamma_c$ fused to an N-terminal nonfluorescent GFP variants mXFPm and mXFPe, respectively. These GFP variants were orthogonally labeled with anti-GFP nanobodies 'minimizer' (MI) conjugated with Rho11 and 'enhancer' (EN) conjugated with DY647 (*Kirchhofer et al., 2010*; *Figure 1C*). Receptor dimerization was quantified by assessing co-tracking individual Rho11 and DY647 molecules. Under these conditions, IL-2Rβ and $\gamma_c$ show minimal co-diffusion in the absence of ligand (*Figure 1D–E*), indicating that the receptors are monomeric. Addition of H9 caused an increase in co-diffusion, consistent with ligand-induced dimerization (*Figure 1D–E*). As predicted, H9-REH and H9-REK showed reduced receptor dimerization relative to H9 as measured by co-tracking of IL-2Rβ and γc (*Figure 1E*). To confirm that reduced receptor dimerization was the result of impaired interactions with $\gamma_c$, we used surface plasmon resonance (SPR) to measure binding of IL-2Rβ and IL-2 muteins to immobilized $\gamma_c$ (*Figure 1F*). Precomplexed cytokine and IL-2Rβ were used in these assays as the affinity of IL-2 for $\gamma_c$ is too weak to measure directly. H9-REH, H9-REK, and H9-RETR had reduced $\gamma_c$ binding relative to H9, indicating the reduced receptor dimerization observed in cells is due to impaired interactions with $\gamma_c$ (*Figure 1G*).

For preliminary characterization, IL-2 variants with enhanced affinity for IL-2Rβ were used to facilitate comparison of partial agonists; however, this strategy may direct activity of IL-2 muteins toward NK cells in vivo due to their high levels of IL-2Rβ expression. To eliminate this potential source of bias, we expressed $\gamma_c$ binding muteins in the context of wild-type human IL-2 (IL-2-REH, IL-2-REK, and IL-2-RETR) and confirmed their effects on STAT5 signaling compared to wild-type IL-2 (*Figure 1—figure supplement 1*).

### The IL-2 receptor partial agonist IL-2-REH specifically increases the frequency and number of regulatory T cells in vivo

To determine the cell type specificity of IL-2 partial agonists in vivo, we took advantage of the strong proliferative effect of IL-2 by assessing the frequency of immune cell populations following cytokine treatment. 30 μg doses of MSA-fused cytokine were administered to C57BL/6J mice on days 0, 3, and 6 prior to analysis on day 7. Under these conditions, we did not observe overt toxicity from

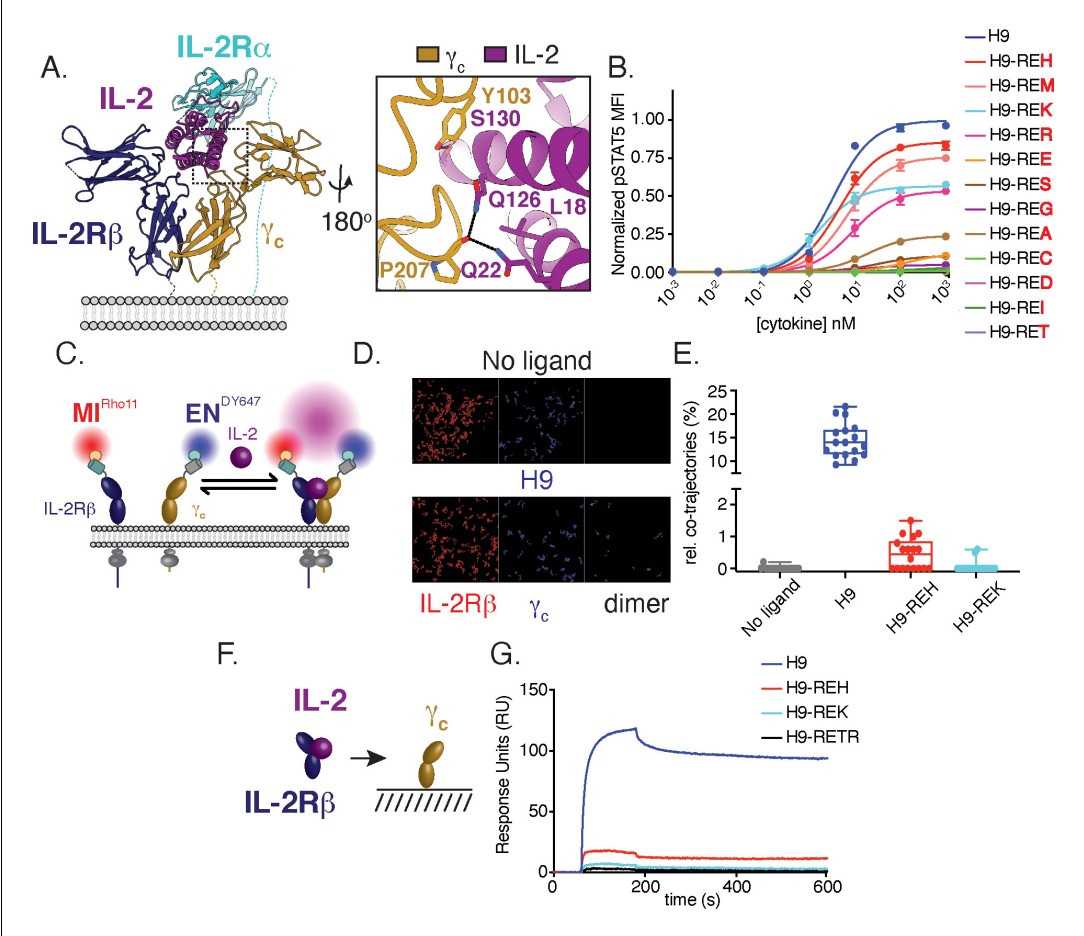

**Figure 1.** Structure-based design and biophysical characterization of IL-2 receptor partial agonists. (**A**) Cartoon representation of the human interleukin-2 (IL-2) receptor complex (PDB: 2B5I). IL-2 is shown in magenta, IL-2 receptor beta (IL-2Rβ) is shown in blue, common gamma (γc) is shown in gold, IL-2 receptor alpha (IL-2Rα) is shown in teal. Insert shows IL-2 residues targeted for mutagenesis. (**B**) Amino acid substitution at IL-2 Q126 tune cytokine activity. IL-2 variants were expressed on the super-2 H9 background (L80F, R81D, L85V, I86V, I92F), an IL-2 variant with high affinity for IL-2Rβ, and assayed for dose-dependent phosphorylation of STAT5 (pSTAT5) by phosphoflow cytometry in YT cells following a 15 min stimulation. MFI, mean fluorescence intensity. Data are shown as mean ± standard deviation of duplicate wells and are representative of two independent experiments. (**C–E**) Mutations in IL-2 at the γc binding interface impair receptor dimerization in the context of live cell membranes. (**C**) Schematic of single-molecule particle tracking experiment. HeLa cells expressing IL-2Rβ and γc with an N-terminal mXFPm and mXFPe were labeled with anti-GFP nanobodies 'minimizer' (MI) and 'enhancer' (EN), conjugated to Rho11 and DY647, respectively. Cells were stimulated with 100 nM H9 or variants as indicated and dual-color single-molecule images were captured by total internal reflection fluorescence microscopy (TIRFM). (**D**) Representative fields showing single-molecule trajectories of Rho11, DY647 and co-trajectories. (**E**) Quantification of IL-2Rβ and γc heterodimerization via the fraction of co-trajectories of Rho11 and DY647. Each dot represents a single cell. (**F–G**) IL-2 partial agonists have impaired binding to γc in solution. (**F**) Schematic of surface plasmon resonance (SPR) binding experiment. (**G**) SPR curves showing binding of pre-complexed human IL-2Rβ with H9 or variants (1:1) to human γc immobilized on a streptavidin sensor chip. Measurements were made with 5 μM analyte and reference subtracted for binding to an irrelevant biotinylated target. RU = response unit.

The online version of this article includes the following figure supplement(s) for figure 1:

**Figure supplement 1.** IL-2 variants with reduced binding to γc modulate effective concentration 50 (EC50) and maximal responsiveness (Emax) of pSTAT5 signaling in the context of human IL-2.

cytokine treatment although IL-2 and to a lesser extent IL-2-REH had increased spleen size relative to mice receiving phosphate-buffered saline (PBS) control (*Figure 2—figure supplement 1A–B*).

IL-2 administration reduced the frequency of T and B cells while causing an increase in the frequency of NK cells and granulocytes (*Figure 2A–D*). Analysis of absolute cell numbers indicated that these changes are due to the expansion of NK cells and granulocytes with no changes in the absolute numbers of T and B cells (*Figure 2—figure supplement 1C–F*). While granulocytes are not

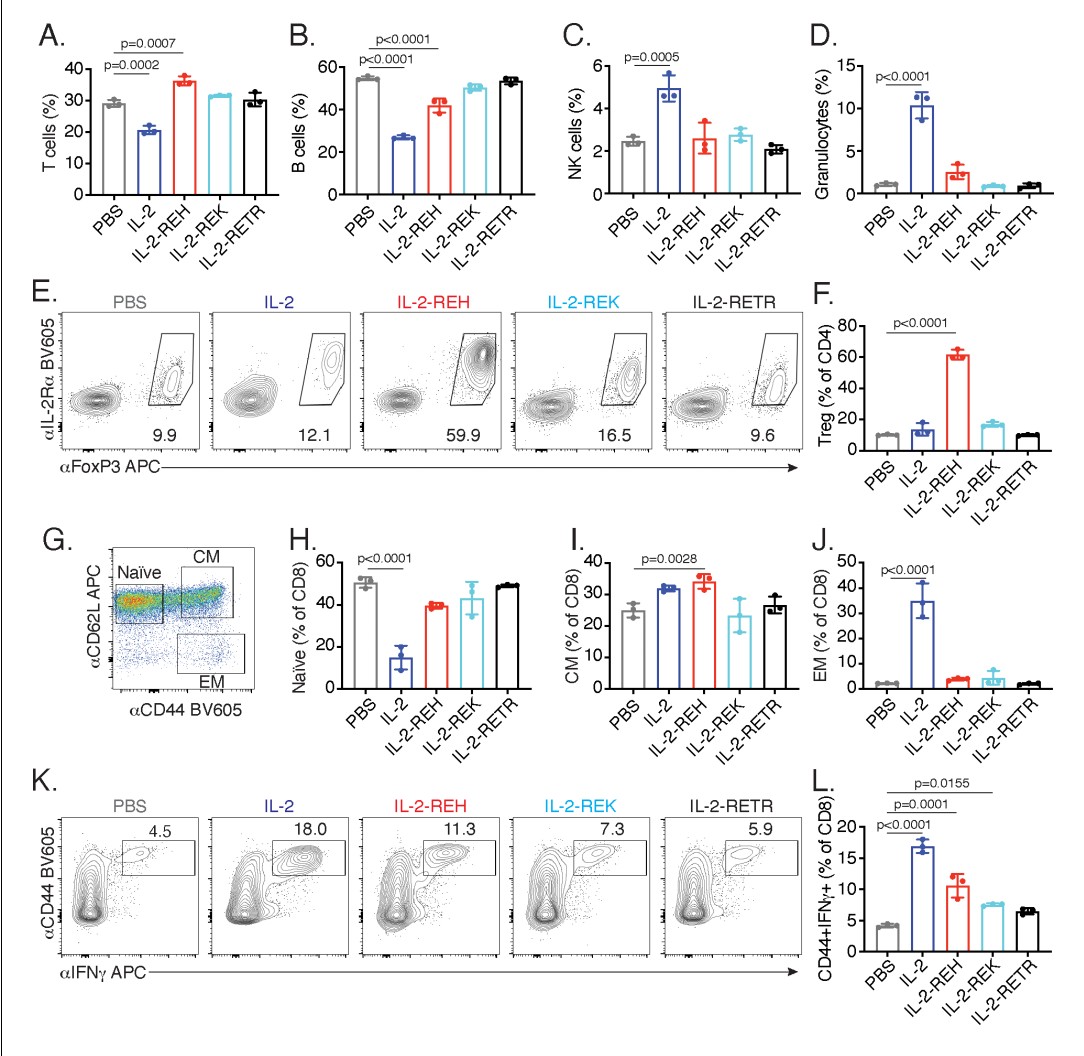

**Figure 2.** IL-2 receptor partial agonists elicit cell type-specific responses in vivo. C57BL/6J mice were dosed with three 30 µg intraperitoneal injections of mouse serum albumin (MSA)-conjugated IL-2 or muteins diluted in PBS on day 0, 3, and 6. On day 7, spleen and lymph nodes were harvested for immune cell profiling. (A-D) IL-2 treatment decreases the frequency of T and B cells while increasing the frequency of NK cells and granulocytes in spleen. T cells were gated as CD3+, B cells were defined as CD3-CD19+, NK cells were analyzed as CD3-NK1.1+, and granulocytes were defined as CD3-Ly6G+SSC$^{hi}$. All populations were expressed as a percentage of live CD45+ cells. For absolute numbers and gating, see *Figure 2—figure supplement 1*. (E) IL-2-REH increases the frequency of regulatory T cells in spleen. Tregs were defined as IL-2Rα+Foxp3+ and expressed as a percentage of liveCD3+CD4+. Similar results were observed in lymph nodes. For absolute numbers, see *Figure 2—figure supplement 2A*. For gating, see *Figure 2—figure supplement 2B*. (G-J) IL-2 increases the frequency of effector memory CD8+ T cells with a decrease in naïve CD8+ T cells while IL-2-REH increases the frequency of central memory CD8+ T cells. (G) Gating scheme to identify naïve, central memory (CM), and effector memory (EM) CD8+ T cells. (H-J) Quantification of memory cell frequency in spleen expressed as a percent of liveCD3+CD8+ T cells. For absolute numbers, see *Figure 2—figure supplement 3A–C*. For gating, see *Figure 2—figure supplement 3D*. (K-L) IL-2, IL-2-REH, and IL-2-REK enhance IFNγ producing CD8+ T cells in proportion to their agonist activity. For cytokine recall, lymph node cells were restimulated with phorbol 12-myristate 13-acetate (PMA) and ionomycin in the presence of Monensin (GolgiStop) for 4 hr. For gating, see *Figure 2—figure supplement 3E*. Bar graphs show mean ± standard deviation of n = 3 mice per group. Data were analyzed by one-way ANOVA with Tukey's multiple comparisons and are representative of two or more independent experiments.

The online version of this article includes the following figure supplement(s) for figure 2:

**Figure supplement 1.** In vivo characterization of IL-2 partial agonists.

**Figure supplement 2.** Quantification of CD4+ T cells following cytokine administration.

**Figure supplement 3.** Analysis of CD8+ T cell phenotype and IFNγ production following cytokine treatment.

thought to be directly responsive to IL-2, the increase in granulocyte frequency and number is likely due to activation or recruitment due to pro-inflammatory mediators produced by NK cells and T cells. Importantly, this effect was attenuated in mice which received IL-2-REH. In contrast, IL-2-REH increased the frequency of T cells while modestly reducing B cell frequency due to expansion of T cells (*Figure 2A–D*, *Figure 2—figure supplement 1C–F*).

The increase in T cell frequency and number following IL-2-REH administration led us to investigate which subpopulation of T cells might be responsible for this effect. Within the CD4+ T cell compartment, Tregs are known to be dependent on IL-2 signaling (*Fontenot et al., 2005*). IL-2 administration did not significantly impact Treg frequency or absolute number; however, IL-2-REH dramatically increased the frequency and number of Tregs without increasing the frequency or number of IL-2Rα+ conventional CD4+ T cells (*Figure 2E–F*, *Figure 2—figure supplement 2*).

In CD8+ T cells, IL-2 is known to drive differentiation to an 'effector memory' phenotype characterized by increased cytotoxicity, expression of the memory T cell marker CD44, and loss of the lymph node homing marker CD62L (*Manjunath et al., 2001*; *Moogk et al., 2016*). In contrast, IL-15, which shares IL-2Rβ/γ$_c$ signaling receptors with IL-2, promotes induction of a 'central memory' phenotype characterized by expression of both CD44 and CD62L. Central memory CD8+ T cells have improved persistence and antitumor effects in the context of adoptive cell therapy (*Klebanoff et al., 2005*; *Manjunath et al., 2001*; *Wherry et al., 2003*). Assessment of CD8+ T cell memory populations following cytokine treatment revealed that IL-2 decreased the frequency of naïve CD8+ T cells with an increase in effector memory T cells while IL-2-REH caused a slight increase in the frequency and number of central memory CD8+ T cells (*Figure 2G–J*, *Figure 2—figure supplement 3A–D*). In addition to altering memory phenotype, IL-2 signaling in CD8+ T cells can induce the expression of the pro-inflammatory cytokine IFNγ which acts as an important mediator of antitumor and antiviral immunity (*Castro et al., 2018*; *Sa et al., 2013*). IL-2, and to a lesser extent IL-2-REH and IL-2-REK, increased the frequency of IFNγ-producing CD8+ T cells following restimulation (*Figure 2K–L*, *Figure 2—figure supplement 3E*). Thus, IL-2-REH preferentially increased Tregs without inducing the increases in NK cells, granulocytes, and CD8+ effector memory T cells observed with IL-2 administration.

## REH increases Treg frequency via selective proliferation of Foxp3+ cells

The increase in regulatory T cell frequency and number following IL-2-REH treatment led us to investigate the source of Tregs in IL-2-REH-treated mice. Regulatory T cells can be broadly classified into two groups, 'natural' Tregs (nTreg) which develop in the thymus and induced Tregs (iTreg) which differentiate from naïve T cells in the periphery (*Curotto de Lafaille and Lafaille, 2009*). The increase in regulatory T cells in response to IL-2-REH could either be due to differentiation of Tregs from conventional CD4+ T cells or proliferation of a preexisting population of Foxp3+ cells. To determine if IL-2-REH caused differentiation of conventional CD4+ T cells into Tregs, we adoptively transferred conventional CD4+ T cells into congenic recipient mice, administered IL-2-REH, and assessed the frequency of Tregs in the donor and recipient compartments (*Figure 3A*). If IL-2-REH induces the differentiation of conventional CD4+ T cells to Tregs, we would expect to find increased Tregs in both the donor and recipient. Alternatively, if IL-2-REH causes the proliferation of a preexisting population of Foxp3+ cells, we would expect to find elevated Treg frequencies in recipient CD4+ T cells but not cells from the donor. IL-2-REH increased the frequency of Tregs in the CD45.1 recipient compartment but not the CD45.2 donor, strongly suggesting that IL-2-REH acts to expand preexisting Tregs (*Figure 3B–C*, *Figure 3—figure supplement 1A*).

Given that Tregs in IL-2-REH-treated mice do not arise through differentiation from conventional CD4+ T cells, the increase may be due to selective proliferation of Tregs in IL-2-REH-treated mice. To test whether IL-2-REH caused specific expansion of Tregs, mice were administered a single injection of cytokine followed by daily administration of the thymidine analog, EdU, to track proliferation (*Figure 3D*). IL-2 and IL-2-REH resulted in a similar level of EdU incorporation in Tregs while IL-2 led to higher levels of EdU incorporation than IL-2-REH in CD8+ T cells (*Figure 3E–F*, *Figure 3—figure supplement 1B*). Together, these results indicate that IL-2-REH selectively expands Tregs in vivo from a preexisting population of Foxp3+ cells.

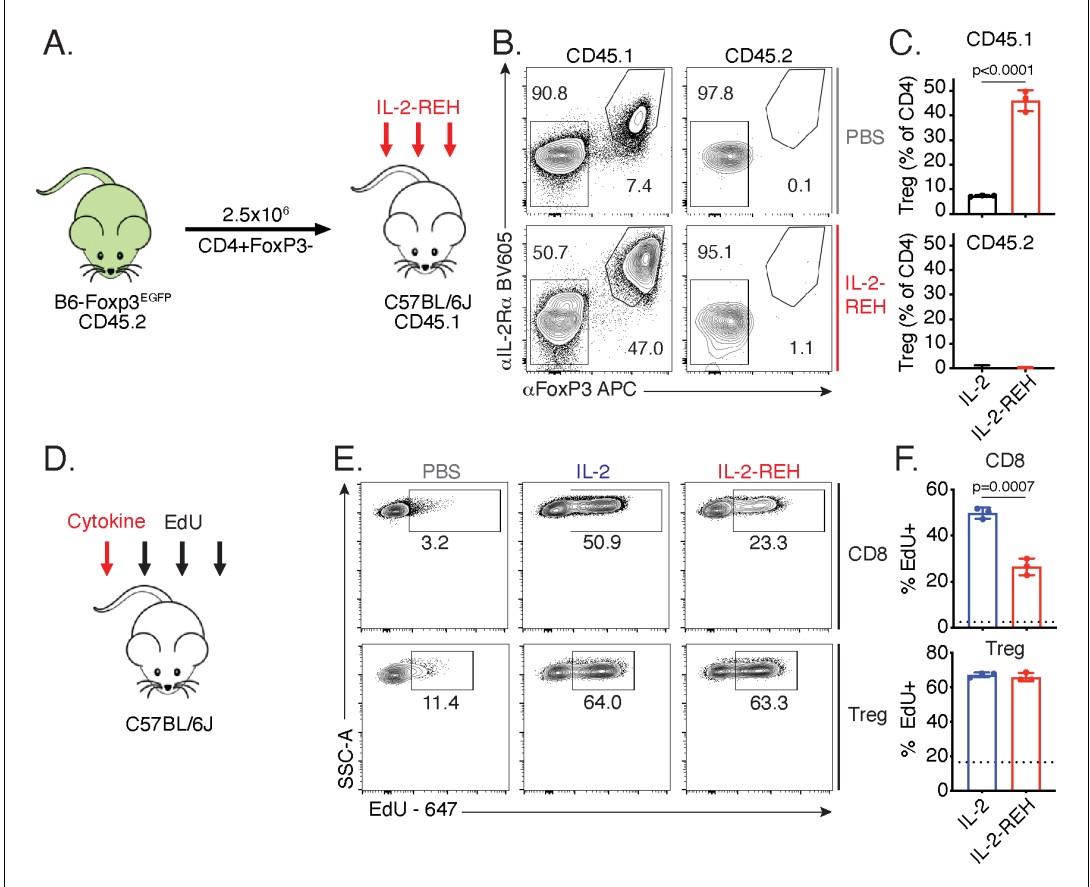

**Figure 3.** IL-2-REH increases Treg frequency via selective proliferation of Foxp3+ cells. (A–C) Increased frequency of Tregs in IL-2-REH-treated mice is not the result of differentiation from conventional CD4+ T cells. (A) Schematic of experimental design. Conventional CD4+ T cells from B6-Foxp3$^{EGFP}$ mice (CD45.2) were transferred to congenic recipients (CD45.1). Mice were treated with three 30 µg injections of IL-2-REH or PBS control on day 0, 3, and 6 prior to analysis on day 7. Flow cytometry plots (B) and quantification (C) of Treg frequencies. Tregs were defined as IL-2Rα+Foxp3+ and expressed as a percent of CD3+CD4+ T cells. For gating, see *Figure 3—figure supplement 1A*. (D–F) IL-2-REH induces proliferation of Tregs with reduced activity on CD8+ T cells relative to IL-2. (D) Schematic of proliferation tracking experiment. C57BL/6J mice were treated with a single 30 µg injection of cytokine on day 0 followed by EdU injections on day 0, 1, and 2 prior to analysis on day 3. Flow cytometry plots (E) and quantification (F) of EdU incorporation in Tregs (CD3+CD4+Foxp3+) and CD8+ T cells (CD3+CD8+). Dashed line indicates EdU incorporation in PBS-treated mice. For gating, see *Figure 3—figure supplement 1B*. Bar graphs show mean ± standard deviation of n = 3 mice per group. Groups were compared by unpaired t-test and data are representative of two independent experiments.

The online version of this article includes the following figure supplement(s) for figure 3:

**Figure supplement 1.** Representative gating schemes for adoptive transfer and proliferation tracking experiments.

## IL-2-REH promotes greater IL-2 signaling in Tregs than CD8+ T cells

The ability of IL-2-REH to preferentially support Treg proliferation in vivo could be due to cell type-intrinsic differences in IL-2 sensitivity between CD8+ T cells and Tregs or due to indirect effects from multiple IL-2 responsive populations in vivo. To determine if IL-2 and IL-2-REH elicit distinct signaling in Tregs and CD8+ T cells, we analyzed STAT5 phosphorylation and transcriptional changes following cytokine stimulation ex vivo. In regulatory T cells, IL-2-REH induced phosphorylation of STAT5 to about 40% of the maximal level achieved by IL-2 following a 20 min stimulation (*Figure 4A*). RNA-seq analysis revealed that IL-2 and IL-2-REH induced very similar gene expression programs in Tregs following a 4 hr cytokine stimulation as indicated by the small number of differentially regulated genes (*Figure 4B*). This was not due to lack of response as we identified many genes that were differentially regulated following stimulation with IL-2 compared to unstimulated Tregs, including the GPCR chaperone, *Ramp3,* and the metabolic enzyme, glutaminase 2 (*Gls2*) (*Figure 4C*). A well-described role of IL-2 in T cells is to promote survival and proliferation while enhancing effector

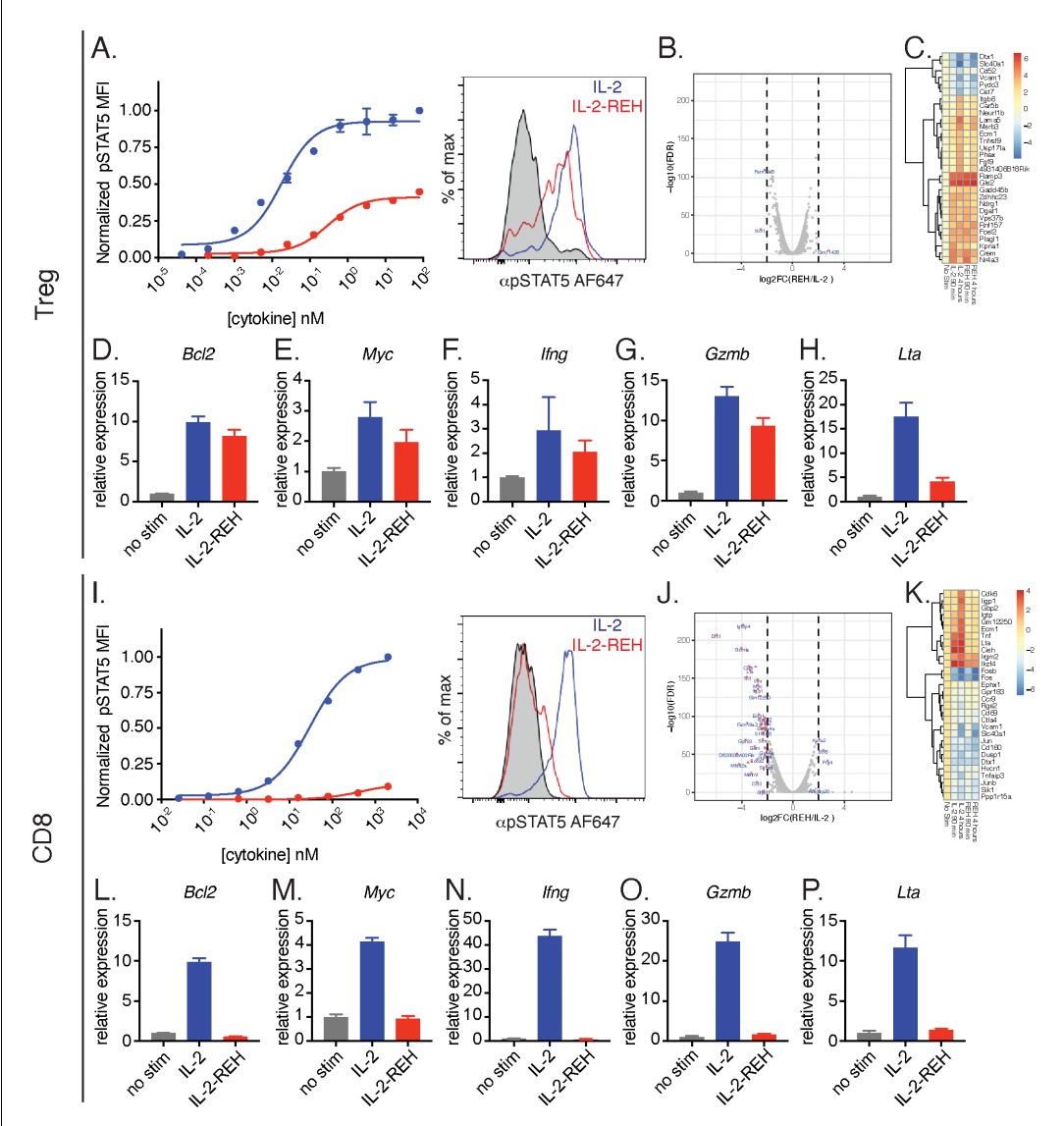

**Figure 4.** IL-2-REH selectively promotes signaling in Tregs with reduced activity on CD8+ T cells relative to IL-2. (**A**) IL-2-REH promotes STAT5 phosphorylation in regulatory T cells. Dose-response curve (left) and histogram at 80 nM (right) showing STAT5 phosphorylation in Tregs following a 20 min cytokine stimulation. Dose-response curves show mean ± standard deviation of duplicate wells. MFI, mean fluorescence intensity. (**B–C**) IL-2 and IL-2-REH produce similar transcriptional profiles in regulatory T cells. Purified Tregs from B6-Foxp3$^{EGFP}$ mice were stimulated with 200 nM cytokine for 90 min or 4 hr prior to RNA-seq. (**B**) Volcano plot of differentially expressed genes in Tregs following 4 hr stimulation with IL-2 or IL-2-REH. (**C**) Heatmap showing the top 30 differentially regulated genes (sorted by FDR) between unstimulated and IL-2-treated Tregs. Expression was normalized to that of unstimulated Tregs to show cytokine-induced changes. (**D–H**) Quantitative polymerase chain reaction (qPCR) of IL-2 regulated genes in Tregs following a 4 hr stimulation with 200 nM cytokine. For quantification, cycle threshold (Ct) values were normalized to expression of *Gapdh* and expressed as fold change relative to unstimulated cells. Bar graphs show mean ± standard deviation of triplicate measurements. (**I**) IL-2-REH fails to induce STAT5 phosphorylation in CD8+ T cells. Dose-response (left) and histogram at 2 μM (right) of STAT5 phosphorylation in ex vivo CD8+ T cells following a 20 min cytokine stimulation. Dose-response curves show mean ± standard deviation of duplicate wells. (**J–K**) IL-2 and IL-2-REH induce distinct transcriptional profiles in CD8+ T cells. Purified CD8+ T cells were stimulated with 200 nM cytokine for 90 min or 4 hr prior to RNA-seq. (**J**) Volcano plot showing differentially expressed genes in CD8+ T cells following 4 hr stimulation with IL-2 or IL-2-REH. (**K**) Heatmap showing the top 30 differentially regulated genes between unstimulated and IL-2-treated CD8+ T cells. Expression was normalized to that of unstimulated CD8+ T cells. (**L–P**) qPCR of IL-2 regulated genes in CD8+ T cells following a 4 hr stimulation with 200 nM cytokine. Data were analyzed as in D–H. Results are representative of two or more independent experiments.

functions including cytokine secretion. To assess these effects, we performed quantitative polymer-ase chain reaction (qPCR) on a subset of IL-2 regulated genes including the STAT5 target genes, lymphotoxin α (*Lta*) and *Bcl2* (*Villarino et al., 2016*). IL-2 and IL-2-REH increased expression of *Bcl2, Myc, Ifng, Gzmb,* and *Lta* (*Figure 4D–H*); however, quantitative differences between genes were observed with IL-2-REH inducing lower levels of *Lta* relative to IL-2 (*Figure 4H*).

In CD8+ T cells, IL-2 induced STAT5 phosphorylation to a much greater extent than IL-2-REH (*Figure 4I*). Accordingly, RNA-seq analysis identified many differentially regulated genes following stimulation with IL-2 or IL-2-REH (*Figure 4J–K*). Among the genes upregulated by IL-2 were the cell cycle gene *Cdk6* and the effector cytokines *Tnf* and *Lta*. These transcriptional changes were con-firmed by qPCR where IL-2 but not IL-2-REH resulted in upregulation of *Bcl2, Myc, Ifng*, *Gzmb*, and *Lta* (*Figure 4L–P*). The differences between IL-2-REH responsiveness in vitro suggest that in vivo dif-ferences in cytokine-induced proliferation may reflect intrinsic differences in cytokine responsiveness between cell subsets with IL-2-REH eliciting preferential signaling in Tregs relative to CD8+ T cells.

## IL-2-REH exploits intrinsic differences in IL-2 signaling to elicit cell type-specific activity

To extend our observations to a broader set of IL-2 responsive lymphocytes, we profiled IL-2 recep-tor expression and pSTAT5 signaling in NK cells and T cells following a 20 min cytokine stimulation (*Figure 5—figure supplement 1*). NK cells, which are negative for IL-2Rα but express high levels of IL-2Rβ and intermediate levels of $\gamma_c$, respond selectively to IL-2 but not IL-2-REH, IL-2-REK, or, IL-2-RETR. Similarly, ex vivo CD4+ and CD8+ T cells, which are negative for IL-2Rα and express low to intermediate levels of IL-2Rβ and $\gamma_c$, respond to IL-2 but not partial agonists. In contrast, Tregs which constitutively express IL-2Rα and intermediate levels of IL-2Rβ and $\gamma_c$ respond to IL-2 and IL-2-REH but only weakly to IL-2-REK. Activated CD4+ and CD8+ T cells (blasts) express elevated levels of IL-2Rα, IL-2Rβ, and $\gamma_c$, have increased responsiveness to both IL-2-REH and IL-2-REK, with IL-2-REH act-ing as a full agonist on CD8+ T cell blasts. Thus, cells differ greatly in their responsiveness to IL-2 partial agonist based on cell type and activation status.

To better understand the relationship between IL-2 receptor expression and sensitivity to partial agonists, we plotted STAT5 phosphorylation against receptor expression for each of the cell types analyzed. IL-2-REH and IL-2-REK responsiveness positively correlated with expression of IL-2Rα and $\gamma_c$ but not IL-2Rβ, indicating that expression of IL-2Rα and $\gamma_c$ may contribute to the selectivity of IL-2 partial agonists (*Figure 5A–C*).

Given the positive correlation between IL-2Rα expression and responsiveness to partial agonists, we sought to test if IL-2Rα dependence was important for the selective activity of IL-2-REH in vivo. To do so, we installed the IL-2-REH $\gamma_c$ binding mutations in the context of super-2 (H9), an IL-2 vari-ant with high affinity for IL-2Rβ and *reduced* dependence on IL-2Rα (*Levin et al., 2012*). Increased affinity for IL-2Rβ (H9-REH) attenuated the IL-2-REH-mediated increase in Treg frequency, indicating that IL-2Rα dependence is important for cell type selectivity of IL-2-REH in vivo (*Figure 5D–E*).

To directly assess the role of IL-2Rα expression in IL-2-REH responsiveness, we generated an IL-2Rα deficient variant of CTLL-2s, an IL-2-dependent murine cell line, by CRISPR knockout. Wild-type and IL-2Rα-deficient cells were co-cultured at a 1:1 ratio in IL-2 or IL-2-REH for 72 hr, and changes in the frequency of IL-2Rα-expressing cells were used as a readout of IL-2Rα dependence. When cul-tured in IL-2, cells remained roughly at their input ratios, suggesting a negligible contribution of IL-2Rα expression. In contrast, co-culture in IL-2-REH produced an enrichment in IL-2Rα-expressing cells suggesting knockout cells were reduced in their capacity to respond to IL-2-REH (*Figure 5F–G*).

In addition to constitutive expression of IL-2Rα, Tregs are further sensitized to IL-2 signaling by miR-155 dependent suppression of SOCS1 (*Lu et al., 2015*; *Lu et al., 2009*). As SOCS1 acts as a negative regulator of IL-2 signaling by inhibiting JAK1 (*Liau et al., 2018*), we reasoned that SOCS1 expression may disproportionately inhibit signaling in response to IL-2-REH, which induces lower lev-els of receptor dimerization relative to wild-type IL-2. To determine the impact of SOCS1 on STAT5 signaling, we transduced CTLL-2 cells with lentivirus encoding SOCS1 along with a GFP expression marker. Wild-type and SOCS1 overexpressing cells were mixed, stimulated with IL-2 or IL-2-REH, and the ratio of pSTAT5 signaling in SOCS1 expressing cells relative to wild-type CTLL-2s was quan-tified. Overexpression of SOCS1 impaired phosphorylation of STAT5 in response to IL-2-REH but not IL-2, suggesting that the IL-2-REH signaling is more sensitive to negative regulation by SOCS1

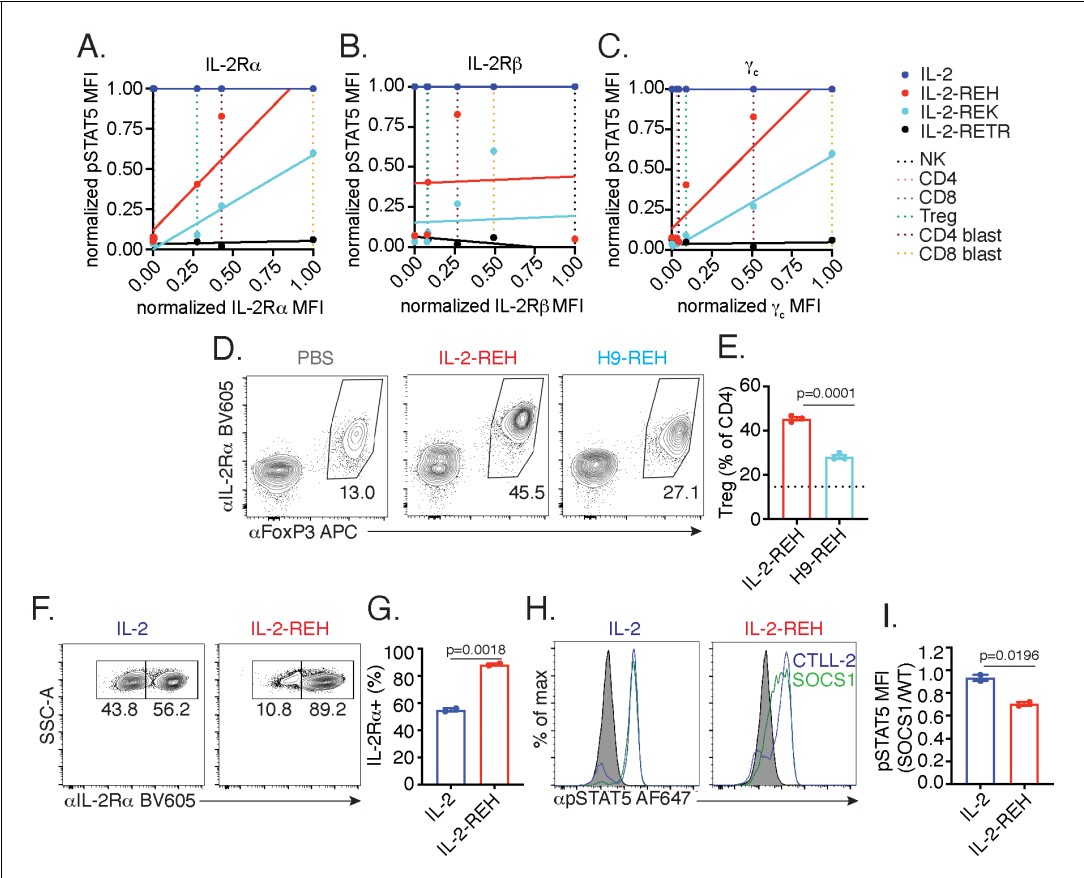

**Figure 5.** IL-2-REH exploits intrinsic differences in IL-2 signaling to elicit cell type-specific activity. (A–C) IL-2 partial agonist activity depends on IL-2Rα and γc expression. Correlation plots showing STAT5 phosphorylation and receptor expression in NK cells, CD4+ T cells, CD8+ T cells, and Tregs. Dashed lines represent receptor expression normalized to the highest and lowest expressing cell types. pSTAT5 MFI at saturating cytokine concentration was normalized to stimulation with IL-2. See also *Figure 5—figure supplement 1*. (D–E) IL-2Rβ affinity regulates cell type selectivity of IL-2-REH. C57BL/6J mice were dosed with three 30 μg intraperitoneal injections of IL-2-REH or H9-REH diluted in PBS on day 0, 3, and 6 followed by analysis on day 7. Flow cytometry plots (D) and quantification (E) of Treg frequency expressed as a percent of CD3+CD4+ T cells. Bar graphs show mean ± standard deviation of n = 3 mice per group. Dashed line represents Treg frequency in PBS-treated mice. Groups were compared using an unpaired t-test. (F–G) IL-2Rα expression controls proliferation in response to IL-2-REH. Wild-type and IL-2Rα$^{-/-}$ CTLL-2 cells were mixed at 1:1 ratio and cultured in the presence of 200 nM IL-2 or IL-2-REH for three days prior to quantification of IL-2Rα expression by flow cytometry. Bar graph shows mean ± standard deviation of duplicate wells. Groups were compared using an unpaired t-test. (H–I) IL-2-REH signaling is more sensitive to negative regulation by suppressor of cytokine signaling 1 (SOCS1). SOCS1-p2a-eGFP expressing CTLL-2s were mixed with wild-type CTLL-2 cells, rested overnight, and stimulated with 1 μM IL-2 or IL-2-REH for 30 min prior to fixation, permeabilization, and detection of pSTAT5 by phosphoflow cytometry. Wild-type and SOCS1 overexpressing cells were gated on the basis of GFP expression. (H) Histograms showing pSTAT5 staining in wild-type (blue) and SOCS1 expressing (green) CTLL-2 cells. (I) pSTAT5 staining in SOCS1 expressing cells expressed as a ratio to pSTAT5 staining in wild-type CTLL-2s. Bar graph shows mean ± standard deviation of duplicate wells. Groups were compared using an unpaired t-test. MFI, mean fluorescence intensity. Results are representative of at least two independent experiments.

The online version of this article includes the following figure supplement(s) for figure 5:

**Figure supplement 1.** IL-2 receptor staining and signaling in NK cells and T cells.

(*Figure 5H–I*). Thus, IL-2-REH exploits two key features of IL-2 sensitivity in Tregs, constitutive IL-2Rα expression and miR-155 suppression of SOCS1.

## IL-2-REH expanded Tregs retain suppressive capacity and enhance recovery from DSS-induced colitis

We hypothesized that IL-2-REH-mediated expansion of Tregs might have utility in the suppression of immune activation in the setting of autoimmune disease. To confirm that expanded Tregs retained their ability to suppress T cell proliferation following IL-2-REH treatment, we tracked Treg expansion

over time and performed Treg suppression assays using Tregs from PBS, IL-2, or IL-2-REH-treated mice. Tregs from cytokine-treated mice were capable of suppressing proliferation of conventional CD4+ T cells (Tcon) at similar ratios, indicating that IL-2-REH-induced proliferation does not impair the suppressive capacity of expanded Tregs (*Figure 6—figure supplement 1*).

An early demonstration of the suppressive capacity of regulatory T cells was provided by adoptive transfer experiments showing that Tregs are able to suppress T cell-mediated colitis (*Read et al., 2000*). In this model, colitis is induced by transfer of naïve CD45RB^{hi} cells into immunodeficient mice and can be prevented by co-transfer of CD45RB^{lo} cells of which IL-2Rα+Tregs are essential (*Ostanin et al., 2009*). To determine if IL-2-REH could expand Tregs and prevent disease in the context of T cell colitis, we transferred CD45RB^{lo} cells with or without Tregs and administered IL-2-REH at the onset of weight loss (*Figure 6—figure supplement 2A*). Under these conditions, Tregs were able to suppress colitis; however, this was inhibited by administration of IL-2-REH (*Figure 6—figure supplement 2B*). Interestingly, this was not due to an inability of IL-2-REH to expand Tregs as elevated Treg frequency was observed in cytokine-treated mice (*Figure 6—figure supplement 2C*). Similarly, IL-2-REH was able to stimulate Tregs as measured by expression of activation markers IL-2Rα, GzmB, and ICOS (*Figure 6—figure supplement 2D–F*). These findings suggest that Treg expansion at later time points does not correlate with disease protection in the transfer model of T cell colitis; however, an important caveat is that the high frequency of responding T cells in this case may not reflect the typical circumstance during inflammatory disease progression. Furthermore, the IL-2Rα-selectivity of IL-2-REH may simultaneously activate IL-2Rα+CD8+ T cells in inflammatory lesion, providing an opposing effect to therapeutic benefit from the Treg expansion.

To determine if systemic treatment with IL-2-REH could provide therapeutic benefit during an ongoing immune response, we used the DSS-induced colitis model, treating mice with IL-2, IL-2-REH, or the previously described IL-2Rα selective IL-2 mutein, IL-2-N88D (*Peterson et al., 2018*), starting on day 3 at the onset of DSS-induced weight loss (*Figure 6A*). In these experiments, we administered lower doses of IL-2 similar to those used in the treatment of nonobese diabetic mice when adjusted for the molecular weight of the fusion protein (*Grinberg-Bleyer et al., 2010*). All mice lost weight as a result of DSS administration; however, IL-2-REH improved recovery as measured by body weight on day 16 (*Figure 6B*, *Figure 6—figure supplement 3*). This effect was unique to IL-2-REH and not seen in mice receiving PBS control, IL-2, or, IL-2-N88D which lost significant amounts of weight relative to untreated mice. Consistent with these findings, hematoxylin and eosin (H and E) staining of distal colon sections showed elevated disease score following DSS induction, an effect which was attenuated in mice receiving IL-2-REH but not IL-2 or IL-2-N88D (*Figure 6C–D*). The mitigating effects of IL-2-REH administration on weight loss and disease score were accompanied by increased Treg frequency in the mesenteric lymph nodes, indicating that IL-2-REH expansion of Tregs is maintained in the context of an inflammatory response (*Figure 6E*). Importantly, DSS induction did not alter IL-2-REH specificity as IL-2 but not IL-2-REH or IL-2-N88D increased NK cell frequency in the mesenteric lymph node (*Figure 6F*).

To further characterize the effect of cytokine treatment on disease progression, we performed immunohistochemistry of colon sections from control and diseased mice. While disease induction did not impact Treg frequency in mesenteric lymph nodes, DSS administration enhanced Treg infiltration in the colon, an effect that was not significantly impacted by cytokine treatment (*Figure 6G*). Previously, the cytotoxic effector molecule, granzyme B (GzmB) has been implicated in the suppressive activity of Tregs through killing of antigen presenting cells (*Cao et al., 2007*). Treatment with IL-2, IL-2-REH, or, IL-2-N88D increased the presence GzmB+Foxp3+ cells, suggesting that all of these cytokines were capable of promoting Treg activation in the tissue (*Figure 6H*); however, in the case of IL-2 and IL-2-N88D, this did not translate to improved outcome. Thus, IL-2-REH retains the ability to expand and activate Tregs in the context of an ongoing immune response, thereby improving recovery from DSS-induced colitis.

The discrepancy between our findings in T cell transfer colitis and DSS-induced colitis suggest that the timing and nature of the inflammatory response play a large role in the efficacy of Treg expansion in ameliorating pathological inflammation. These findings highlight the complexity of strategies for Treg expansion as a therapeutic regimen in autoimmune disease.

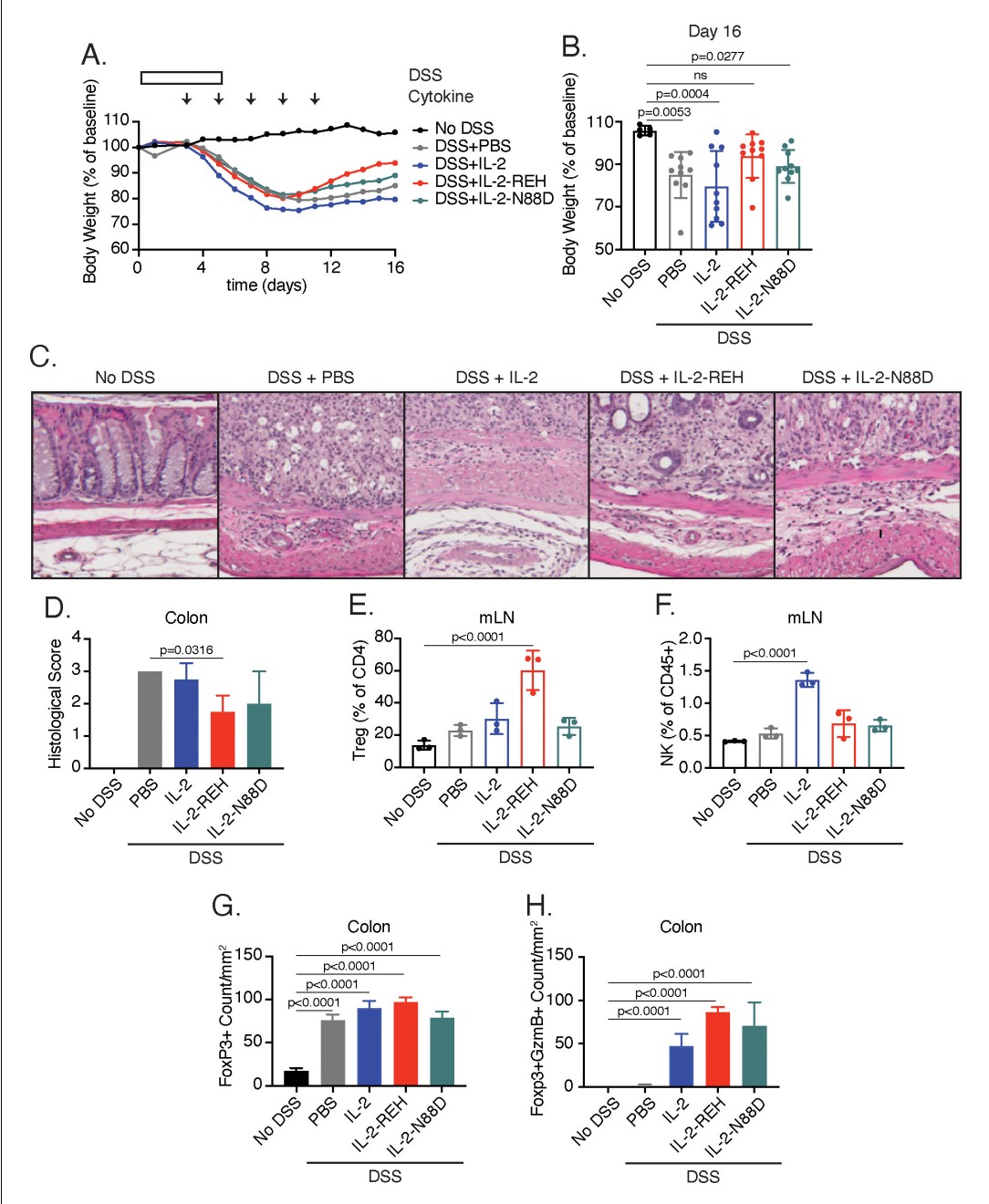

**Figure 6.** IL-2-REH enhances recovery from DSS-induced colitis. C57BL/6J mice were placed on 3–3.5% dextran sulfate sodium (DSS in drinking water for 6 days. Starting on day 3, mice received subcutaneous injections of PBS (n = 10), 5 µg IL-2 (n = 10), 5 µg IL-2-REH (n = 10) or 5.3 µg IL-2-N88D (n = 10) every other day for 10 days. Data were compared to mice maintained on drinking water without DSS (n = 5). (**A**) Mean body weight measurements following 3% DSS induction. Weights were normalized to body weight on day 0. For body weight curves of individual mice, see *Figure 6—figure supplement 3A–E*. (**B**) IL-2-REH enhances recovery from DSS-induced colitis. Comparison of body weights on day 16. If mice died prior to day 16, the final body weight measurement was used for analysis. For survival curve, see *Figure 6—figure supplement 3F*. Bar graphs show mean ± standard deviation and were analyzed by one-way ANOVA relative to no DSS condition, multiple comparisons were corrected using Dunnett's test. Data are representative of two independent experiments. (**C-D**) IL-2-REH attenuates DSS-induced colon damage. (**C**) Representative hematoxylin and eosin (H and E) staining of distal colon sections on day 13. (**D**) Quantification of disease score from H&E staining of distal colon. Mice received no DSS (n = 3) or 3.5% DSS with PBS (n = 4), 5 µg IL-2 (n = 4), 5 µg IL-2-REH (n = 4), or 5.3 µg IL-2-N88D (n = 3). Groups were compared by one-way ANOVA with Tukey's multiple comparisons. (**E**) IL-2-REH treatment increases the frequency of regulatory T cells in mesenteric lymph nodes (mLNs). Flow cytometry analysis of mLN cellularity was performed 10 days post 3% DSS induction. Bar graphs show mean ± standard deviation of n = 3 mice per group. Data were analyzed by one-way ANOVA with Tukey's multiple comparisons. (**F**) IL-2 increases NK cell frequency in the mesenteric

*Figure 6 continued on next page*

*Figure 6 continued*

lymph nodes, analyzed as in E. (**G-H**) Quantification of colon infiltrating Tregs by immunohistochemistry (IHC). Bar graphs show mean ± standard deviation of colon sections from n = 3 mice per group. (**G**) DSS increases Treg infiltration in colon. Quantification of Tregs by IHC staining for Foxp3 in colon. (**H**) Cytokine treatment enhances Treg activation in colon. Quantification of Foxp3+GzmB+ cells by IHC. Groups were compared using one-way ANOVA with Tukey's multiple comparisons relative to untreated mice.

The online version of this article includes the following figure supplement(s) for figure 6:

**Figure supplement 1.** In vitro suppression and kinetic analysis of IL-2-REH expanded Tregs.
**Figure supplement 2.** Effects of IL-2-REH on T cell transfer colitis.
**Figure supplement 3.** Weight loss and survival in DSS-induced colitis.

## Discussion

IL-2 play a central role in the initiation, maintenance, and resolution of the immune response by activating multiple cell types. The countervailing effects of IL-2 to both activate and suppress the immune response is critical for maintaining immune homeostasis but has hampered basic understanding of IL-2 activity on different cell types in vivo and limited the clinical utility of cytokine administration. Here, we identify an IL-2 agonist which specifically expands a preexisting population of regulatory T cells due to differences in IL-2 receptor expression and signaling pathway intermediates. The mechanistic principles we demonstrate here for determining cell type selectivity of IL-2 partial agonists can likely be generalized to other cytokine and growth factor systems that signal through dimeric receptors.

This mechanism-based approach taken here differs from previously reported IL-2 muteins with attenuated binding to IL-2Rβ and thus increased dependence on IL-2Rα (*Khoryati et al., 2020*; *Peterson et al., 2018*; *Shanafelt et al., 2000*). Here, the strategy of systematically altering the affinity of IL-2 for $\gamma_c$ to generate IL-2 variants with submaximal signaling at saturating concentrations of cytokine is conceptually analogous to modulation of G-protein coupled receptor (GPCR) signaling by partial agonists (*Pliska, 1999*; *Shukla, 2019*). Unlike GPCR which undergo ligand-induced confirmational changes, cytokine receptor signaling depends largely on receptor dimerization (*Engelowski et al., 2018*). We find that modulating the affinity of IL-2 for its receptor can shift the monomer-dimer equilibrium and promote distinct levels of intracellular signaling. Importantly, these effects are highly dependent on expression of receptor subunits such that IL-2 muteins which fail to signal on one cell population act as full or partial agonists on others (*Figure 5—figure supplement 1*). The affinity of IL-2 for IL-2Rβ predominantly impacts the dose sensitivity of cytokine signaling rather than the maximal response such increased cytokine concentration can compensate for reduced IL-2Rβ affinity. The approach we have taken here in which we modulate the interaction between IL-2 and $\gamma_c$ is less dependent on cytokine concentration and has the capacity to alter receptor dimerization even at saturating concentrations of cytokine (*Figure 1E*).

Cell type-specific signaling through IL-2Rβ/$\gamma_c$ is not unique to engineered cytokines. Indeed, IL-2 shares signaling receptors IL-2Rβ and $\gamma_c$ with IL-15 (*Carson et al., 1994*; *Giri et al., 1994*). Despite exhibiting very similar signaling on CD8+ T cells in vitro (*Ring et al., 2012*), IL-2 and IL-15 have many divergent biological functions in vivo (*Lodolce et al., 1998*; *Ma et al., 2006*). Differences in IL-2 and IL-15 have been attributed to unique receptors, IL-2Rα and IL-15Rα, which play important roles in determining cell type specificity of cytokine signaling (*Waldmann, 2006*). In contrast to IL-2 and IL-15, partial agonists achieve cell type selectivity through a distinct mechanism, relying on different thresholds for IL-2 signaling across cell types rather than unique alpha receptors to achieve biological specificity.

This mechanism of functional diversification through altered receptor dimerization shares some features with type I interferons (IFNs), a large class of cytokines that signal though receptors IFNAR1 and IFNAR2. Despite the use of a shared receptor complex, type I IFNs play nonredundant roles in antiviral immunity by modulating the strength of receptor dimerization (*Ng et al., 2016*; *Ng et al., 2015*). While IL-2 and IL-15 are the only natural ligands of the IL-2Rβ/$\gamma_c$ receptor complex, we show here that additional IL-2 receptor agonists with divergent biological behavior can be generated by systematically altering receptor binding. This strategy relies on intrinsic differences in cytokine sensitivity that can be exploited by designed cytokine variants to satisfy thresholds on certain cell types but not others. As many cytokines use differences in receptor expression and signaling pathway

components to achieve cell type selectivity, this approach may be more generally applicable to other pleiotropic cytokines.

## Materials and methods

### Protein expression and purification

Constructs encoding mouse serum albumin (MSA, amino acids 19–608) fused to wild-type or mutant human IL-2 (amino acids 21–153) were cloned into the pAcGP67a vector (BD Biosciences) with an N-terminal gp67 signal peptide and a C-terminal 6xHis tag. The super-2 variant H9 (L80F, R81D, L85V, I86V, I92F) with enhanced affinity for IL-2Rβ was used as indicated (*Levin et al., 2012*). All mutations are listed based on their position in the mature polypeptide for consistency with previous reports. For binding studies, the extracellular domains of IL2Rβ (amino acids 27–240) and γ$_c$ (amino acids 23–254) with a C-terminal biotin acceptor peptide (N-GLNDIFEAQKIEWHE-C) were cloned into the pAcGP67a vector (BD Biosciences) with an N-terminal gp67 signal peptide and a C-terminal 6xHis tag. Baculovirus was produced by transfection of *Spodotera frugiperda* (Sf9) insect cells with Cellfectin II (Gibco) and Sapphire Baculovirus DNA (Allele) followed by viral amplification in Sf9. High titer virus was used to infect *Trichoplusia ni* (High-Five) cells (Invitrogen). Protein was isolated from supernatant 48–72 hr post infection by Ni-NTA affinity chromatography and size exclusion chromatography (SEC) using Superdex S200 column (GE). Purity was confirmed by SDS-PAGE and IL-2 bioactivity was tested by pSTAT5 signaling on YT cells. For biotinylation, human γ$_c$ was expressed, purified by Ni-NTA affinity chromatography, and incubated with soluble BirA ligase in buffer containing 0.5 mM Bicine (pH 8.3), 100 mM ATP, 100 mM magnesium acetate, and 500 µM biotin overnight at 4C (*Fairhead and Howarth, 2015*). Biotinylated γ$_c$ was purified by SEC as described above.

For in vivo experiments, endotoxin was removed using High Capacity NoEndo Columns (Viva Products) and levels were quantified using the LAL Chromogenic Endotoxin Quantitation Kit (Thermo). All in vivo preparations had <1U endotoxin/dose. For DSS studies, the previously described IL-2 mutein with reduced affinity for IL-2Rβ, IL-2-N88D, was expressed with T3A mutation to eliminate O-glycosylation and C125A mutation to prevent intrachain disulfides. To extend half-life, N88D was expressed with an N-terminal Fc domain lacking Fc effector functions (hIgG1-P329G LALA) (*Schlothauer et al., 2016*) in Expi293 cells (Gibco) by transient transfection.

### Single-molecule particle tracking

Receptor dimerization in the plasma membrane was quantified by dual-color single-molecule co-tracking as described in detail previously (*Mendoza et al., 2019*; *Wilmes et al., 2015*). For this purpose, HeLa cells were transfected with IL-2Rβ and γ$_c$ N-terminally fused to engineered, nonfluorescent variants of monomeric GFP mXFPm and mXFPe, respectively, that are orthogonally recognized by the anti-GFP nanobodies 'Minimizer' (MI) and 'Enhancer' (EN) (*Kirchhofer et al., 2010*). After transfection, cells were seeded on coverslips coated with PLL-PEG-RGD (*You et al., 2014*) and imaging experiments were performed 48 hr post transfection. Single-molecule imaging experiments were conducted by total internal reflection fluorescence (TIRF) microscopy with an inverted microscope (Olympus IX71) equipped with a triple-line total internal reflection (TIR) illumination condenser (Olympus). A 150× magnification objective with a numerical aperture of 1.45 (UAPO 150 ×/1.45 TIRFM, Olympus) was used for TIR illumination of the sample. All experiments were carried out at room temperature in medium without phenol red, supplemented with an oxygen scavenger and a redox-active photoprotectant to minimize photobleaching (*Vogelsang et al., 2008*). Cell surface receptors were labeled by addition of 2.5 nM of each MI and EN conjugated to Rho11 (ATTO-TEC) and DY647 (Dyomics), respectively, which were kept in the supernatant throughout imaging experiments. After incubation with IL-2 variants for 5 min, dual-color videos of 150 frames were recorded at a rate of 31.25 Hz using total internal reflection fluorescence microscopy. The fluorophores were simultaneously excited at 561 nm and 642 nm and fluorescent signals of the respective channels were separated with an image splitter and detected with a backilluminated electron multiplied (EM) CCD camera (iXon DU897D, Andor Technology). Single molecule tracking and co-tracking was performed using 'SLIMfast,' an in-house software written in MATLAB (kindly provided by Christian P. Richter) based on the multiple-target tracing (MTT) algorithm (*Sergé et al., 2008*). Individual

receptor dimers were identified by co-localization within a maximum intermolecular distance of 100 nm and a minimum trajectory length of 10 consecutive frames (*Wilmes et al., 2015*).

## Surface plasmon resonance

Interactions between IL-2 muteins and $\gamma_c$ were assessed by surface plasmon resonance (SPR) on a BIAcore T100 instrument. Biotinylated human $\gamma_c$ was immobilized on a streptavidin sensor chip (GE) at low density (<100 RU). To facilitate measurement of complex assembly, IL-2 muteins were expressed in the context of H9 and incubated with equimolar human IL-2Rβ. Binding measurements were performed at 25°C with a flow rate of 50 μL/min. Precomplexed cytokine and receptor (5 μM) were injected over the chip for 120 s followed by a 500 s dissociation phase. Following each injection, the chip was regenerated with 4M $MgCl_2$. Traces were background-subtracted for binding to an unrelated biotinylated protein immobilized in the reference channel.

## Cell culture

Mammalian cell lines were maintained in complete RPMI (RPMI 1640-GlutaMAX supplemented with 10% fetal bovine serum, nonessential amino acids, sodium pyruvate, 15 mM HEPES and penicillin-streptomycin) at 37°C with 5% $CO_2$. Cells were tested for mycoplasma using MycoAltert PLUS mycoplasma detection kit (Lonza). YT cells were maintained at low passage number and verified by IL-2 receptor expression and responsiveness (*Yodoi et al., 1985*). CTLL-2 cells (AATC TIB-214), a murine IL-2-dependent T cell line, were obtained from the vendor and maintained in complete RPMI supplemented with 1000 IU/mL recombinant mouse IL-2. IL-2Rα-deficient CTLL-2 cells were generated by lentiviral transduction using lentiCRISPRv2 encoding gRNA 5'-ACAACTTACCCAGAAATCGG-3' (*Sanjana et al., 2014*). Transduced cells were sorted on the basis of IL-2Rα expression using a SH800S cell sorter equipped with 100 μm chip (Sony) and mixed 1:1 with wild-type CTLL-2 cells. Mixed cells were cultured in the presence of 200 nM IL-2 or IL-2-REH for 3 days prior to analysis of cell counts and IL-2Rα+ cell frequency. For SOCS1 overexpression, the open reading frame encoding mouse SOCS1 (1–212) followed by a p2a ribosomal skip sequence and enhanced green fluorescent protein (eGFP) was cloned into lentiviral gene ontology vector, LeGO (*Weber et al., 2010*). Cells were transduced with lentivirus and sorted on the basis of eGFP expression using a SH800S cell sorter equipped with 100 μm chip (Sony). Transduced cells were mixed 1:1 with wild-type CTLL-2s and stimulated with 1 μM IL-2 or IL-2-REH prior to analysis of STAT phosphorylation by flow cytometry.

## pSTAT5 staining

$2–3 \times 10^5$ cells were stimulated in 100 μL complete RPMI with cytokine in a 96 well plate prior to fixation with 1.6% paraformaldehyde for 10 min at room temperature. Cells were permeabilized with 100% ice-cold methanol and stored at −20°C prior to staining. Cells were washed twice with FACS buffer (PBS pH7.2, 2% FBS, 2 mM EDTA) and stained with 1:50 anti-STAT5 pY694 AF647 (BD) for 1 hr at room temperature. Mean fluorescence intensity (MFI) was monitored using a CytoFLEX flow cytometer (Beckman Coulter). MFI was normalized to pSTAT5 MFI in the absence of cytokine and at saturating levels of IL-2.

For T cell signaling, spleen and lymph nodes from B6-Foxp3$^{EGFP}$ mice (C.Cg-*Foxp3*$^{tm2Tch}$/J) were processed to generate a single cell suspension. Fc receptors were blocked with TrueStain FcX (Biolegend) and cells were stained with αCD3 AF700 (500A2, eBiosience), αCD4 eFluor 450 (GK1.5, eBioscience), and αCD8 BV785 (53–6.7, BioLegend). CD8+ T cells were defined as CD3+CD8+, conventional CD4+ T cells were defined as CD3+CD4+Foxp3, while Tregs were gated as CD3+CD4+Foxp3+ on the basis of GFP expression.

For NK cell signaling, murine NK cells were isolated from spleen and lymph nodes of C57BL/6J mice by negative selection using Miltenyi NK cell isolation kit (Miltenyi Biotec). To enhance recovery, purified NK cells were mixed with CellTrace Violet (Invitrogen) labeled carrier cells and stimulated for 20 min at 37°C. Cells were fixed using Lyse/Fix buffer (BD) and permeabilized using Phosflow Perm Buffer III (BD).

For IL-2 receptor staining, a single cell suspension from spleen and lymph nodes of B6-Foxp3$^{EGFP}$ mice was pre-stained with TrueStain FcX (Biolegend) followed by a panel of αCD3 PerCP-Cyanine5.5 (145–2 C11, eBioscience), αCD4 eFluor 450 (GK1.5, eBioscience), αCD8 BV785 (53–6.7, BioLegend),

αNK1.1 PE (PK136, eBioscience), and αCD19 PC7 (1D3, eBioscience). To this panel were added antibodies against IL-2Rα (CD25, PC61, Biolegend), IL-2Rβ (CD122, TM-b1, Biolegend), or $\gamma_c$ (CD132, TUGm2, Biolegend) labeled in APC.

## RNA-seq and qPCR

Tregs were isolated from B6-Foxp3[EGFP] mice using CD4+ T cell isolation kit (Miltenyi) followed by fluorescence-activated cell sorting (FACS) for GFP expression to isolate Tregs. CD8+ T cells were isolated from C57BL/6J mice using CD8+ T cell isolation kit (Miltenyi). Isolated cells were stimulated with 200 nM IL-2 or IL-2-REH and RNA was extracted using the Zymo RNA miniprep kit (Zymo Research). 500 ng RNA was used for RNA-Seq library preparation with the Kapa mRNA HyperPrep Kit (KK8580, Kapa Biosystems) and indexed with NEXTflex DNA Barcodes-24 as previously reported (Lin et al., 2017). Indexed samples were loaded to 2% E-Gel pre-cast gels (ThermoFisher). DNA fragments between 250 bp and 400 bp were recovered and purified with Zymoclean Gel DNA Recovery Kit (ThermoFisher), quantified by Qubit (Invitrogen), and sequenced on an Illumina HiSeq3000 system. Sequenced reads (50 bp, single end) were obtained with the Illumina CASAVA pipeline and mapped to the mouse genome mm10 (GRCm38, Dec. 2011) using Bowtie2 and Tophat2. Only uniquely mapped reads were retained. Raw counts that fell on exons of each gene were calculated and normalized by using RPKM (Reads Per Kilobase per Million mapped reads). Differentially expressed genes were identified with the R Bioconductor package 'edgeR' and expression heat maps were generated with the R package 'pheatmap'.

For quantitative polymerase chain reaction (qPCR), Tregs and CD8+ T cells were stimulated for 4 hr with 200 nM cytokine prior to RNA isolation using RNeasy kit (QIAGEN). cDNA was generated using the iScript reverse transcriptase kit. qPCR was performed using PowerTrack SYBR Green (ThermoFisher) and analyzed on StepOnePlus Real-Time PCR System (Applied Biosystems). For analysis, Ct values were exported from StepOne Software and ΔCt was calculated for all samples relative to *Gapdh* control. ΔΔCt was calculated relative to unstimulated cells and fold change was expressed as $2^{-\Delta\Delta Ct}$.

## Cell profiling

C57BL/6J mice were maintained in specific pathogen-free housing under a protocol approved by the Stanford Veterinary Service Center. For profiling experiments, mice were administered 30 μg injections of IL-2 or muteins on days 0, 3, and 6. On day 7, spleen and lymph nodes were harvested and a single cell suspension was generated. For transcription factor staining, cells were surface stained prior to fixation and permeabilization using Foxp3 transcription factor buffer set (eBiosciences). For cytokine recall, cells were cultured in 1 μg/mL Ionomycin (Sigma), 50 ng/mL phorbol 12-myristate 13-acetate (PMA, Sigma) and GolgiStop (BD) for 4 hr. Cells were surface stained prior to fixation and permeabilization using Cytofix/Cytoperm kit (BD).

## Adoptive transfer

Conventional CD4+ T cells from B6-Foxp3[EGFP] were isolated using the CD4+ T cell isolation kit (Miltenyi) followed by FACS of conventional T cells (GFP negative) cells using a SH800S cell sorter equipped with 100 μm chip (Sony). Isolated cells were resuspended in PBS and transferred via retro-orbital injection into CD45.1 mice (B6.SJL-*Ptprc*[a]*Pepc*[b]/BoyJ) and allowed to engraft for 4 hr prior to injection of 30 μg IL-2-REH or PBS control. Mice were administered two additional cytokine doses on days 3 and 6 prior to analysis of Treg frequencies on day 7.

## In vivo proliferation tracking

C57BL/6J mice were administered a single 30 μg dose of IL-2 or IL-2-REH followed by daily 100 μg injections of Click-iT EdU (Invitrogen) on day 0, 1, and 2. On day 3, spleen and lymph nodes were harvested. Cells were surface stained, fixed, and permeabilized using Foxp3 transcription factor buffer set (eBiosciences). Following Foxp3 staining, cells were washed and resuspended in Click-iT saponin based permeabilization and wash reagent prior to performing click reaction according to manufactures protocols.

## In vitro Treg suppression

CD4+ T cells from B6-Foxp3$^{EGFP}$ mice were isolated by magnetic-activated cell sorting (MACS) followed by FACS for GFP+ (Treg) and GFP– (Tcon) cells using a SH800S cell sorter equipped with 100 µm chip (Sony). $2.5 \times 10^4$ Tcons from PBS-treated mice were loaded with CellTrace Violet (CTV, Invitrogen) and incubated with 1:1 Dynabeads Mouse T-Activator CD3/CD28 (Gibco) and varying numbers of Tregs from cytokine-treated mice. Cells were cultured for 3 days prior to FACS analysis of CTV dilution in Tcons.

## T cell colitis

CD4+ T cells were isolated from the spleens of BALB/c mice (Taconic) by MACS followed by FACS sorting for CD45RB$^{hi}$CD25- (naïve) and CD45RB$^{lo}$CD25+ (Treg). $3 \times 10^5$ naïve CD4+ T cells were transferred with or without $1 \times 10^4$ Tregs into female C. B-17 scid mice (Taconic). Body weight was monitored weekly and cytokine administration was initiated at the onset of weight loss on day 24. Mice received 5 µg injections of IL-2-REH twice weekly for three weeks followed by a second week-long course beginning on day 57. The frequency and activation of Tregs were monitored in blood on day 42.

## DSS-induced colitis

Female C57BL/6 mice (B6NTac, Taconic) aged 8–12 weeks were administered 3–3.5% dextran sulfate sodium (DSS) (36,000–50,000 M.Wt, MP Biomedical) in drinking water ad libitum for 6 days. Starting on day 3 of DSS treatment, mice were injected with cytokine or PBS control subcutaneously (s.c.) every other day for a total of five injections. For flow cytometry and histological analysis, mice were euthanized by $CO_2$ asphyxiation on day 10. Colons were removed, flushed, and 2–2.5 cm pieces of the most distal part of the colon were fixed in 10% buffered formalin for histology and IHC analysis. Histological scoring for tissue damage was performed according to *Wirtz et al., 2017*.

## Statistics

Statistics were performed using Prism v8.4 (GraphPad Software). Data are expressed as mean ± standard deviation unless otherwise indicated.

# Acknowledgements

The authors would like to thank Geoff Smith and Arthur Weiss for technical guidance and thoughtful discussion. Emma Dean, Casey Weaver, Amy Fan, Robert Saxton and members of the Garcia Lab provided critical feedback and thoughtful discussion.

# Additional information

#### Competing interests

Caleb R Glassman, Leon Su, Sonia S Majri-Morrison: Coauthor on a patent which includes discoveries described in this manuscript (WO/2019/104092). K Christopher Garcia: Coauthor on a patent which includes discoveries described in this manuscript (WO/2019/104092). Founder of Synthekine. The other authors declare that no competing interests exist.

#### Funding

| Funder | Grant reference number | Author |
| --- | --- | --- |
| National Institute of Allergy and Infectious Diseases | R37-AI051321 | K Christopher Garcia |
| Czech Science Foundation | 18-12973S | Marek Kovar |
| National Science Foundation | DGE-1656518 | Caleb R Glassman |

The funders had no role in study design, data collection and interpretation, or the decision to submit the work for publication.

## Author contributions
Caleb R Glassman, Formal analysis, Investigation, Methodology, Writing - original draft, Writing - review and editing; Leon Su, Sonia S Majri-Morrison, Magdiel Pérez-Cruz, Peggy P Ho, Ievgen Koliesnik, Nadine Nagy, Tereza Hnizdilova, Lora K Picton, Investigation, Methodology, Writing - review and editing; Hauke Winkelmann, Fei Mo, Peng Li, Formal analysis, Investigation, Methodology, Writing - review and editing; Marek Kovar, Paul Bollyky, Lawrence Steinman, Jacob Piehler, Warren J Leonard, Methodology, Project administration, Writing - review and editing; Everett Meyer, Methodology, Writing - review and editing; K Christopher Garcia, Conceptualization, Funding acquisition, Writing - original draft, Project administration, Writing - review and editing

## Author ORCIDs
Caleb R Glassman (ID) https://orcid.org/0000-0002-3342-7989
Leon Su (ID) http://orcid.org/0000-0001-7654-9997
Sonia S Majri-Morrison (ID) http://orcid.org/0000-0003-2612-328X
Hauke Winkelmann (ID) http://orcid.org/0000-0003-3688-6854
Marek Kovar (ID) http://orcid.org/0000-0002-6602-1678
Paul Bollyky (ID) https://orcid.org/0000-0003-2499-9448
Jacob Piehler (ID) http://orcid.org/0000-0002-2143-2270
Warren J Leonard (ID) http://orcid.org/0000-0002-5740-7448
K Christopher Garcia (ID) https://orcid.org/0000-0001-9273-0278

## Ethics
Animal experimentation: Mice were maintained in the Stanford animal facility according to protocols approved by the Stanford University Institutional Animal Care and Use Committee under Administrative Panel on Laboratory Animal Care (APLAC) protocol number 32279.

## Decision letter and Author response
Decision letter https://doi.org/10.7554/eLife.65777.sa1
Author response https://doi.org/10.7554/eLife.65777.sa2

# Additional files

## Supplementary files
• Transparent reporting form

## Data availability
Sequencing data have been deposited in GEO under accession number, GSE162928.

The following dataset was generated:

| Author(s) | Year | Dataset title | Dataset URL | Database and Identifier |
|---|---|---|---|---|
| Glassman CR, Su L, Majri-Morrison SS, Winkelmann H, Mo F, Li P, Pérez-Cruz M, Ho PP, Koliesnik L, Nagy N, Hnizdilova T, Picton L, Kovář M, Bollyky P, Steinman L, Meyer E, Piehler J, Leonard WJ, Garica KC | 2020 | Calibration of cell-intrinsic Interleukin-2 response thresholds guides design of a regulatory T cell biased agonist | https://www.ncbi.nlm.nih.gov/geo/query/acc.cgi?acc=GSE162928 | NCBI Gene Expression Omnibus, GSE162928 |

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

# Appendix 1

**Appendix 1—key resources table**

| Reagent type (species) or resource | Designation | Source or reference | Identifiers | Additional information |
|---|---|---|---|---|
| Gene (*M. musculus*) | Serum Albumin | Uniprot | P07724 | |
| Gene (*Homo-sapiens*) | Interleukin 2 (IL-2) | Uniprot | P60568 | |
| Gene (*Homo-sapiens*) | Interleukin two receptor beta (IL-2Rβ) | Uniprot | P14784 | |
| Gene (*Homo-sapiens*) | Common gamma chain (γ$_c$) | Uniprot | P31785 | |
| Gene (*M. musculus*) | Suppressor of cytokine signaling 1 (SOCS1) | Uniprot | O35716 | |
| Strain, strain background (*M. musculus*) | Female C57BL/6J | The Jackson Laboratory | JAX:000664 | |
| Strain, strain background (*M. musculus*) | Female B6.SJL-*Ptprc$^a$ Pepc$^b$*/BoyJ | The Jackson Laboratory | JAX:002014 | |
| Strain, strain background (*M. musculus*) | Female B6.Cg-*Foxp3$^{tm2Tch}$*/J | The Jackson Laboratory | JAX:006772 | |
| Strain, strain background (*M. musculus*) | Female C57BL/6 | Taconic | C57BL/6NTac | |
| Strain, strain background (*M. musculus*) | Female BALB/c | Taconic | BALB/cAnNTac | |
| Strain, strain background (*M. musculus*) | Female C.B-17 scid | Taconic | C.B-*Igh-1$^b$*/IcrTac-*Prkdc$^{scid}$* | |
| Cell line (*M. musculus*) | Mouse T Cell line (CTLL-2) | ATCC | TIB-214 | |
| Cell line (*Homo-sapiens*) | Human NK Cell line (YT) | *Yodoi et al., 1985* | | |
| Cell line (*Spodoptera frugiperda*) | Insect ovarian Cell line | ATCC | Cat#CRL-1711 | |
| Cell line (*Trichoplusia ni)* | Insect ovarian Cell line | Expression Systems | Cat#94–002F | |
| Cell line (*Homo-sapiens*) | Expi293F | Gibco | Cat#A14527 | |
| Cell line (*Homo-sapiens*) | Human adenocarcinoma Cell line (Hela) | ATCC | Cat#CCL-2 | |
| Antibody | Anti-mouse CD16/32 TruStain FcX (93) rat monoclonal | BioLegend | Cat#101320 | (1:100) |

*Continued on next page*

*Appendix 1—key resources table continued*

| Reagent type (species) or resource | Designation | Source or reference | Identifiers | Additional information |
|---|---|---|---|---|
| Antibody | Anti-mouse CD3ε (145–2 c11) Armenian hamster monoclonal | BioLegend | Cat#100340 | (2.5 µg/mL) plate bound |
| Antibody | Anti-mouse CD28 (37.51) Syrian hamster monoclonal | Bio X Cell | Cat#BE0015-1 | (5 µg/mL) |
| Antibody | Anti-mouse CD45.1 e450 (A20) mouse monoclonal | eBioscience | Cat#48-0453-82 | (1:100) |
| Antibody | Anti-mouse CD45.2 APC-eF780 (104) mouse monoclonal | eBioscience | Cat#47-0454-82 | (1:100) |
| Antibody | Anti-mouse CD45.2 APC (104) mouse monoclonal | eBioscience | Cat#17-0454-82 | (1:100) |
| Antibody | Anti-mouse CD3ε BV785 (17A2) rat monoclonal | BioLegend | Cat#100232 | (1:100) |
| Antibody | Anti-mouse CD3ε AF700 (eBio500A2) Syrian hamster monoclonal | eBioscience | Cat#56-0033-82 | (1:100) |
| Antibody | Anti-mouse CD3ε PerCP-Cy5.5 (145–2 C11) Armenian hamster monoclonal | eBioscience | Cat#44-0031-82 | (1:100) |
| Antibody | Anti-mouse CD19 PE-Cy7 (1D3) rat monoclonal | eBioscience | Cat#25-0193-81 | (1:100) |
| Antibody | Anti-mouse NK1.1 FITC (PK136) mouse monoclonal | eBioscience | Cat#11-5941-85 | (1:100) |
| Antibody | Anti-mouse NK1.1 PE (PK136) mouse monoclonal | eBioscience | Cat#12-5941-82 | (1:100) |
| Antibody | Anti-mouse Ly6G eF450 (RB6-8C5) rat monoclonal | eBioscience | Cat#48-5931-82 | (1:100) |
| Antibody | Anti-mouse CD25 BV605 (PC61) rat monoclonal | BioLegend | Cat# 102035 | (1:100) |
| Antibody | Anti-mouse CD25 APC (PC61) rat monoclonal | BioLegend | Cat#102011 | (1:100) |
| Antibody | Anti-mouse CD122 APC (TM-β1) rat monoclonal | BioLegend | Cat#123214 | (1:100) |
| Antibody | Anti-mouse CD132 APC (TUGm2) rat monoclonal | BioLegend | Cat#132308 | (1:100) |

*Continued on next page*

*Appendix 1—key resources table continued*

| Reagent type (species) or resource | Designation | Source or reference | Identifiers | Additional information |
|---|---|---|---|---|
| Antibody | Anti-mouse Foxp3 APC (FJK-16S) rat monoclonal | eBioscience | Cat#17-5773-82 | (1:100) |
| Antibody | Anti-mouse Foxp3 PE (FJK-16s) rat monoclonal | eBioscience | Cat#12-5773-82 | (1:100) |
| Antibody | Anti-mouse CD4 FITC (RM4.5) rat monoclonal | eBioscience | Cat#11-0042-85 | (1:100) |
| Antibody | Anti-mouse CD4 PE-Cy7 (GK1.5) rat monoclonal | eBioscience | Cat#25-0041-82 | (1:100) |
| Antibody | Anti-mouse CD4 Pacific Blue (GK1.5) rat monoclonal | BioLegend | Cat# 100427 | (1:100) |
| Antibody | Anti-mouse CD8 Alexa Fluor 488 (53–6.7) rat monoclonal | BioLegend | Cat#100726 | (1:100) |
| Antibody | Anti-mouse CD8 BV785 (53–6.7) rat monoclonal | BioLegend | Cat#100750 | (1:100) |
| Antibody | Anti-mouse CD62L (MEL-14) rat monoclonal | BioLegend | Cat# 104411 | (1:100) |
| Antibody | Anti-mouse CD44 (IM7) rat monoclonal | BioLegend | Cat#103047 | (1:100) |
| Antibody | Anti-mouse IFN$\gamma$ AF647 (XMG1.2) rat monoclonal | BD | Cat#557735 | (1:100) |
| Antibody | Anti-pSTAT5 AF647 (47/Stat5 pY694) mouse monoclonal | BD | Cat#612599 | (1:100) |
| Recombinant DNA reagent | LeGO mSOCS1-p2a-eGFP | This study | | Modified from *Weber et al., 2010* mSOCS1 1–212 AS linker p2a cleavage peptide eGFP (UniProt C5MKY7) |
| Recombinant DNA reagent | lentiCRISPR v2 sgIL2RA | This study | | Modified from *Sanjana et al., 2014*, see gRNA seq below |
| Recombinant DNA reagent | pSEMS leader-mXFPm IL-2R$\beta$ | This study | | mECFP (W67F, E143K) hIL-2R$\beta$ 27–551 |
| Recombinant DNA reagent | pSEMS leader-mXFPe $\gamma_c$ | This study | | mEGFP (Y67F, N199D, Y201F) h$\gamma_c$23–369 |
| Sequence-based reagent | mouse IL-2R$\alpha$ gRNA | This study | gRNA | ACAACTTACCCAGAAATCGG |
| Sequence-based reagent | Gapdh_F | IDT | PCR primer | GTGGAGTCATACTGGAACATGTAG |
| Sequence-based reagent | Gapdh_R | IDT | PCR primer | AATGGTGAAGGTCGGTGTG |

*Appendix 1—key resources table continued*

| Reagent type (species) or resource | Designation | Source or reference | Identifiers | Additional information |
|---|---|---|---|---|
| Sequence-based reagent | Bcl2_F | IDT | PCR primer | CCAGGAGAAATCAAACAGAGGT |
| Sequence-based reagent | Bcl2_R | IDT | PCR primer | GATGACTGAGTACCTGAACCG |
| Sequence-based reagent | Ifng_F | IDT | PCR primer | TCCACATCTATGCCACTTGAG |
| Sequence-based reagent | Ifng_R | IDT | PCR primer | CTGAGACAATGAACGCTACACA |
| Sequence-based reagent | Gzmb_F | IDT | PCR primer | CATGTCCCCCGATGATCTC |
| Sequence-based reagent | Gzmb_R | IDT | PCR primer | AAGAGAGCAAGGACAACACTC |
| Sequence-based reagent | Lta_F | IDT | PCR primer | TCTCCAGAGCAGTGAGTTCT |
| Sequence-based reagent | Lta_R | IDT | PCR primer | CTCAGAAGCACTTGACCCAT |
| Sequence-based reagent | Myc_F | IDT | PCR primer | GTCGTAGTCGAGGTCATAGTTC |
| Sequence-based reagent | Myc_R | IDT | PCR primer | CTGTTTGAAGGCTGGATTTCC |
| Peptide, recombinant protein | MSA-IL-2 and variants | This study | | MSA aa19-608, GGGSGS linker, human IL-2 aa21-153, 6xHis |
| Peptide, recombinant protein | MSA-H9 and variants | This study | | MSA aa19-608, GGGSGS linker, human IL-2 (L80F, R81D, L85V, I86V, I92F) aa21-153, 6xHis |
| Peptide, recombinant protein | Fc-IL-2-N88D | This study | | hIgG1-P329G LALA IL-2-N88D (T3A, C125A), 6xHis |
| Peptide, recombinant protein | Anti-GFP minimizer Rho11 | *Kirchhofer et al., 2010* | | |
| Peptide, recombinant protein | Anti-GFP enhancer DY647 | *Kirchhofer et al., 2010* | | |
| Peptide, recombinant protein | Human IL-2Rβ extracellular domain | This study | | aa27-240, 6xHis |
| Peptide, recombinant protein | Human $\gamma_c$ extracellular domain, biotin | This study | | aa23-254, Biotin Acceptor Peptide, 6xHis |
| Commercial assay or kit | CD4[+] T Cell Isolation Kit, mouse | Miltenyi Biotec | Cat#130-104-454 | |

*Continued on next page*

*Appendix 1—key resources table continued*

| Reagent type (species) or resource | Designation | Source or reference | Identifiers | Additional information |
|---|---|---|---|---|
| Commercial assay or kit | Mouse CD8α + T cell isolation kit | Miltenyi Biotec | Cat#130-104-075 | |
| Commercial assay or kit | Mouse NK cell Isolation Kit | Miltenyi Biotec | Cat#130-115-818 | |
| Commercial assay or kit | LS magnetic selection column | Miltenyi Biotec | Cat#130-042-401 | |
| Commercial assay or kit | High Capacity NoEndo Columns | Protein Ark | Cat#Gen-NoE48HC | |
| Commercial assay or kit | Chromogenic Endotoxin Quant Kit | Pierce | Cat#A39552 | |
| Commercial assay or kit | RNeasy Plus Mini Kit | Qiagen | Cat#74134 | |
| Commercial assay or kit | iScript Reverse Transcription Supermix | BioRad | Cat#1708840 | |
| Commercial assay or kit | PowerTrack SYBR Green | Applied Biosystems | Cat#A46109 | |
| Commercial assay or kit | CellTrace Violet Cell Proliferation Kit | Invitrogen | Cat# C34557 | |
| Commercial assay or kit | Click-iT EdU Alexa Fluor 647 Flow Cytometry Assay Kit | Invitrogen | Cat#C10424 | |
| Commercial assay or kit | Propidium Iodide | BD | Cat#556463 | |
| Commercial assay or kit | Fixable Viability Dye eFluor 780 | eBioscience | Cat#65-0865-14 | |
| Commercial assay or kit | Foxp3/Transcription Factor Staining Buffer Set | eBioscience | Cat#00-5523-00 | |
| Commercial assay or kit | Cytofix/Cytoperm Fixation/ Permeablization Kit | BD | Cat#554714 | |
| Commercial assay or kit | GolgiStop | BD | Cat#554724 | |
| Commercial assay or kit | Cellfectin II | Gibco | Cat#10362100 | |
| Commercial assay or kit | Sapphire Baculovirus DNA | Allele | Cat#ABP-BVD-10002 | |
| Commercial assay or kit | FixLyse Buffer | BD | Cat#558049 | |
| Commercial assay or kit | Perm Buffer III | BD | Cat#558050 | |
| Commercial assay or kit | Dynabeads Mouse T-Activator CD3/CD28 for T-Cell Expansion and Activation | Gibco | Cat#11456D | |
| Commercial assay or kit | SA sensor chip | Cytiva | Cat#BR-1005–31 | |

*Continued on next page*

*Appendix 1—key resources table continued*

| Reagent type (species) or resource | Designation | Source or reference | Identifiers | Additional information |
|---|---|---|---|---|
| Commercial assay or kit | Zymo RNA miniprep kit | Zymo Research | Cat#R2051 | |
| Commercial assay or kit | Kapa mRNA HyperPrep Kit | Roche | Cat#08098093702 | |
| Commercial assay or kit | MycoAlert PLUS Mycoplasma Detection Kit (50 Tests) | Lonza | Cat#LT07-705 | |
| Chemical compound, drug | Ionomycin | Sigma | Cat#I9657-1MG | |
| Chemical compound, drug | Phorbol 12-myristate 13-acetate (PMA) | Sigma | Cat#P8139-5MG | |
| Chemical compound, drug | Dextran Sulfate Sodium Salt (36,000–50,000 M.Wt) | MP biomedical | SKU#02160110-CF | |
| Software, algorithm | FlowJo v10.5 | Tree Star | RRID:SCR_008520 | |
| Software, algorithm | GraphPad Prism 8.3.0 | GraphPad Software | RRID:SCR_002798 | |
| Software, algorithm | BIAevaluation software | Cytiva | RRID:SCR_015936 | |
| Software, algorithm | StepOne Software | Applied Biosystems | RRID:SCR_014281 | |
| Software, algorithm | MATLAB | MathWorks | RRID:SCR_001622 | |
| Software, algorithm | Illumina CASAVA pipeline | Illumina | RRID:SCR_001802 | |
| Software, algorithm | Bowtie2 | | RRID:SCR_016368 | |
| Software, algorithm | Tophat2 | | RRID:SCR_013035 | |
| Software, algorithm | edgeR | | RRID:SCR_012802 | |
| Software, algorithm | pheatmap | | RRID:SCR_016418 | |

