## [Decision Letter]

**Acceptance summary:**

You have used structure based design to create IL-2-REH, a partial agonist that can promote preferential expansion of Treg in vivo. Testing IL-2-REH in two models of auto inflammatory disease- the DSS and transfer colitis models- you have found protection in the DSS model, but not the transfer colitis model, despite increase Treg numbers. The results are important to the field in providing a new approach to cytokine engineering and also pointing to the potential complexity of therapies based on Treg expansion in amelioration of auto-inflammatory and autoimmune diseases.

**Decision letter after peer review:**

Thank you for submitting your article "Calibration of cell-intrinsic Interleukin-2 response thresholds guides design of a regulatory T cell biased agonist" for consideration by *eLife*. Your article has been reviewed by 3 peer reviewers, including Michael L Dustin as the Reviewing Editor and Reviewer #1, and the evaluation has been overseen by Tadatsugu Taniguchi as the Senior Editor.

The elegant protein engineering and novelty of the IL-2 weak agonist development is appreciated by all reviewers. However all reviewers concurred that it would be better to test the in vivo function of the IL-2REH expanded Treg and in vivo IL-2REH effects on colitis development in a transfer colitis model, as opposed to the DSS model.

Essential Revisions:

1. You perform a set of transfer colitis experiments to test IL-2REH pre-expanded Treg vs non-expanded Treg,

2. The effect of IL-2REH during the colitis induction by Teff of Teff + Treg.

3. Another essential experiment is to test lower doses of IL-2 as indicated by reviewer 3.

*Reviewer #1:*

Bottom of page 6- the description of the single molecule tracking experiment to investigate receptor dimerisation is not clear. It seems that they have expressed β and common γ chains with the same GFP mutant, but them somehow tagged them with different dye labelled nano bodies and then tracked these on the surface to track movement of heterodimers. The authors need to better explain how β and common γ chains are selectively labelled- there must be some difference in the GFP tags or this would/could not work. The term co-locomotion is unconventional as it suggests that the receptors are self-propelled on the surface of the cell. It should be possible to determine if the motion is diffusive and then co-diffusion. If there is an active component it would more typically be referred to as transport. Locomotion is a term more reserved for movement of cells.

Bottom of page 9- The way the authors refer to recipient and donor seems is written as if it refers to the mice, rather than the recipient (endogenous CD25+ and CD25-) or donor (CD25-) cells. The authors just need to clarify that it’s the cells they are referring to. The logic of the experiment is otherwise sound.

Bottom of page 13- they state that IL-2 REH mirrors two key features of IL-2 sensitivity in Treg. It’s not clear if "mirrors" is the correct term here. Would a better term be "exploits" in selective expansion of Treg in the steady state.

The authors use a DSS colitis model, which is not a classical autoimmune colitis, but more of a transient barrier disruption leading to inflammation that is influenced by T cells, but not directly T cell mediated as transfer colitis model. The in vivo results would be greatly clarified by repeating these points with the transfer colitis model.

*Reviewer #2:*

Many years ago, the laboratory of Chris Garcia embarked into a powerful research line under the general concept of "separating" the different functions of IL-2 from adverse effects as well as, more recently, the differential effects on regulatory T cells (Treg) and effector T cells and NK cells.

For example, some years ago the authors described that IL-2 activity was greater on CD25HI cells (e.g. Treg) in the presence of JES6-1, and greater on CD25LO cells (e.g. CD8^+^ T cells) in the presence of S4B6. JES6-1 and S4B6 are two different anti-IL-2 antibodies.

Chris Garcia's lab also was involved in the generation of several IL-2 muteins, in which the Il2 gene was mutated to improve IL-2 binding to CD122. This generated an IL-2 molecule that was relatively more capable of expanding CTL than Treg, a "superkine" that was, at the time, sought after by cancer immunologists.

In their laboratory's current iteration, Dr. Garcia and co-workers added a number of mutations that weaken the binding to IL2Rg (CD132/Common gc). Their successful structural predictions led to IL-2 muteins that display reduced STAT5 phosphorylation at saturating mutein doses. This manuscript is another useful attempt to understand the differential effects of IL-2, and the practical implications.

Of the shown IL-2 muteins, IL-2-REH is the one with highest phospho-STAT5-inducing activity. It is also the one that induces the highest expansion of Treg cells. Comparatively with IL-2, IL-2-REH is a better inducer of Treg expansion and a poor inducer of CTL expansion and gene expression. Other muteins may have too low an activity on IL-2 receptor to trigger any of the IL-2 functions. The dependence of IL-2-REH on IL2ra for signaling was shown by a number of independent ways, and is convincing.

1. This is not an issue unique to this paper, but it is unclear why IL-2, but, to some degree also IL-2-REH, cause splenomegaly with granulocytosis. Granulocytes don't express IL-2 receptors, so why are they recruited so early and in such high numbers? There must be a very powerful proinflammatory signal produced by NK cells or conventional T cells in response to IL-2. If IL-2-REH is to become a therapy, it would require repeated administration, and, perhaps, it would induce granulocytosis as well in a chronic setting. The readers will benefit from a discussion on this topic.

2. Does IL-2-REH induce IL2ra expression in conventional CD4 cells and CD8 T cells, as does IL-2? The blasts used in Figure 5 are not generated using IL-2 muteins, and IL2ra did not make it to the top 30 genes in Figure 4, so I can't tell. If IL2ra expression is induced by IL-2-REH in conventional T cells, wouldn't one expect that, after a brief delay, IL-2-REH would lead to the proliferation of conventional T cells as well?

3. It is important to characterize the duration of Treg increase (also quality) in vivo, much beyond day 7 post IL-2-REH treatment beginning.

4. From the early Foxp3 days, it was apparent that there is a subset (10-20%) of Foxp3+ Tregs that are CD25-low, in a continuum pattern. In Dr. Garcia's data, both the IL-2 and IL-2-REH expanded Tregs express higher surface IL2ra than the untreated cells or H9-REH-treated cells. Unless the effect is short-lived, the level (MFI) of surface IL2ra expression attained in each case may be correlating with the functionality of the Tregs, and could be another important parameter to consider for suppression of autoimmune disease in vivo.

5. I don't understand the result in Figure 5H, left panel. SOCS1 inhibits IL-2 signaling, but, in this experiment, overexpression of SOCS1 does not appear to have any effect on STAT5 phosphorylation when IL-2 was added. Do CTLL-2 cells express so much intrinsic SOCS1, that overexpression becomes irrelevant? Perhaps the reverse experiment (Socs1 knockdown) should have been attempted. How are the basal levels of SOCS1 in Tconv and Treg, and how does IL-2 and IL-2-REH affects SOCS1 expression?

6. The DSS Colitis experiment is somewhat disappointing. While IL-2-REH drives a striking increase of Treg in mLN and of Foxp3+Gzmb+ cells the colonic tissue, the level of protection from disease is not striking. What other variables could be participating in the amelioration? Perhaps another colitis model, such as the adoptive transfer model (also called the Powrie model), might be beneficial, as it would even test the stability of the transferred Tregs in host mice that did not receive IL-2 or IL-2 muteins.

My conclusion is that this is an important contribution, but some experiments that are easy to perform could add valuable information to the manuscript.

*Reviewer #3:*

First paragraph of the introduction lacks references.

Page 3 Line 14: "severe"

Page 6 line 12: It is not clear what RETR is.

Figure 1 and S1: S1 pSTAT5 MFI shows that human IL-2 saturates the pSTAT5 at 100 nM while H9 (high affinity variant for IL-2Rb) saturates it at 102 nM. This seems to contradict with the justification of the use of H9 instead of WT IL-2. However, towards the end of the Results section H9 is more properly described and referenced as "IL-2 variant with high affinity for IL-2Rb and reduced dependence on IL-2Ra (Levin et al.,2012)". This should come upfront at Figure 1 for clarity. Also, although it is a minor point, if there are TIRF and SPR data for WT IL-2, they can replace the H9 data in Figure 1 or be added to Supplementary Figure 1 as IL-2Ra dependency of REH variants may account for some differences in those readouts too.

Figure 2 and S2: In Figure 2 A-D and S2 B-F, PBS injected spleen percentages add up to more than 100% while the total numbers don't match the T and B percentages. The graphs may come from different experiments, so authors should clarify the mismatches and replace the graphs.

Figure 2 would also benefit from showing a decrease in the total numbers of Foxp3-CD4^+^ effector cells in complementary to the increase in Treg #. IFNγ production of CD4^+^ T cells can also be added to the supplementary figures.

P8 Line 14: "increased cytotoxicity" instead of "increase cytotoxicity"

Figure 3-4-5: Well-performed experiments with clear findings.

Figure 6:

The data presented by the authors are promising, albeit brief. In the colitis model, although IL-2-REH treated mice displayed faster and greater recovery, especially at later time points, findings may be reflective of the IL-2 dose administered. DSS colitis is a chemical injury model that represents the physiology of mucosal healing in human ulcerative colitis, however it is inferior to CD45RBhi CD4^+^ T cell transfer colitis model in the way it represents the contribution of effector and regulatory T cells.

In order to reach a fair conclusion about the therapeutic effect of IL2-REH on colitis, it should also be compared to lower doses of IL-2. The in vitro findings provided in Figure 5 supplement 1 suggest that the negative effects of IL-2 can be alleviated or at least matched to that of IL2-REH at low IL-2 doses. Figure 5 supplement 1 indicates that minimum IL-2 dose (100 nM) that provides maximum pSTAT5 signaling in Tregs doesn't induce signaling in any of the effector cells except for the CD4 blasts. Therefore, a more ideal approach for characterizing the therapeutic effects of IL2-REH in vivo should involve comparison of different IL-2 and IL-2-REH doses in CD45RBhi CD4^+^ T cell transfer colitis model. In such a system, the activity of CD45RBlow CD4^+^ regulatory cells can be tested with or without IL-10 blockade, however blockade experiment is not crucial.

[Editors' note: further revisions were suggested prior to acceptance, as described below.]

Thank you for resubmitting your work entitled "Calibration of cell-intrinsic Interleukin-2 response thresholds guides design of a regulatory T cell biased agonist" for further consideration by *eLife*. Your revised article has been evaluated by Tadatsugu Taniguchi (Senior Editor), a Reviewing Editor, and two of the original reviewers.

You have used structure based design to create IL-2-REH, a partial agonist that can promote preferential expansion of Treg in vivo. Testing IL-2-REH in two models of auto inflammatory disease- the DSS and transfer colitis models- you have found protection in the DSS model, but not the transfer colitis model, despite increase Treg numbers. The results are important to the field in providing a new approach to cytokine engineering and also pointing to limitations of Treg expansion in amelioration of auto inflammatory disease.

The reviewers appreciate your effort to look at the transfer colitis model. The point that IL2REH can promote proliferation, yet not improve the clinical outcome in the model is an important point for the field and should be included in the final paper.

1. Incorporate the transfer colitis data on Treg expansion and clinical scores in a supplemental figure and discuss the implications for the advance in selective Treg expansion, but remaining challenges to translate this into clinical impact.

*Reviewer #1:*

The authors have evaluated IL-2-REH in a different colitis model and tested a range of doses in the DSS model. The results support the ability of IL-2-REH to selectively expand Treg, but the clinical scores in the transfer colitis model were worse with IL-2-REH, even as Treg expansion was increased. The authors have addressed a number of minor issues in the text and added additional discussion that improve the manuscript. The authors have addressed a challenge related to selective expansion of Treg in vivo, but the limitation is that expansion of Treg may not be sufficient, in all cases, to achieve remission of inflammatory disease.

*Reviewer #3:*

While the authors have spent praiseworthy effort on revising the key points raised by reviewers, some points require further clarification and revision. Because of the marginal effects of both IL-2 and IL-2-REH in DSS colitis and slightly disease-exacerbating potential of IL2-REH in the adoptive transfer colitis model, one should be extra cautious to mention the therapeutic effects of this particular IL-2 mutein in colitis. One of the reasons for the unexpected outcome in transfer colitis might actually be due to the timing of initial cytokine dose (day 24) which could have tipped the balance off towards effector activation. Therefore, it would be useful to share transfer colitis data with readers adding also a relevant statement into Discussion section stating the reservations about therapeutic effects of IL-2-REH, at least in the colitis models tested.

I agree that the transfer colitis model, while powerful, is unlikely to show differences between untreated and cytokine treated groups given the data presented in Figure 6—figure supplement 2. However, instead of replacing it with DSS-induced colitis model that only shows some marginal effect and in vitro suppression assays one should rather consider titrating the transferred CD45RBlo/Foxp3.GFP numbers down in transfer colitis setting. This may provide some room for Treg expansion and the effects of cytokine treatment may be visible if a regimen with earlier cytokine doses is adapted. Furthermore, the in vitro suppression assay graph in Figure 6—figure supplement 1 is insufficient without a "No Treg" control to evaluate the degree of suppression in the Treg+ groups.

---

## [Author Response]

Essential Revisions:1. You perform a set of transfer colitis experiments to test IL-2REH pre-expanded Treg vs non-expanded Treg,

We have not carried out this experiment because we have already shown in the paper that IL-2REH expanded Tregs suppress targets as well as IL-2 expanded T cells. In the reviewer’s suggested experiment, T cell transfer colitis, disease is induced by homeostatic proliferation of naïve CD4^+^ T cells, an effect that is suppressed by co-transfer of regulatory T cells. Classic experiments by Fiona Powrie’s group use expression of CD45RB to identify naïve cells (CD45RB^hi^) which are transferred alone or in conjunction with CD45RB^lo^ cells, of which the CD25+ Treg fraction is critical (Read et al., 2000). In our hands, Tregs account for 40-50% of CD45RB^lo^ cells in untreated mice (Author response image 1). Given the abundance of Tregs among CD45RB^lo^ population, we would expect that Tregs from PBS, IL-2 and IL-2-REH all to induce similar levels of disease suppression since, importantly, we have shown that IL-2-REH expanded Tregs retain their suppressive capacity in vitro (Figure 6 —figure supplement 1), but that their suppressive capacity does not exceed that of Treg cells expanded by IL-2.

For these reasons, we feel that T cell transfer colitis experiments using IL-2-REH expanded Tregs will provide limited additional value beyond the data presented in current Figure 6 —figure supplement 1 showing that IL-2-REH expanded Tregs retain suppressive capacity.

**Author response image 1. sa2fig1:** Expression of Foxp3 in CD45RB^hi^ and CD45RB^lo^ cells. CD4^+^ T cells (CD3+CD4^+^) from spleen of B6-Foxp3^EGFP^ mice were gated on CD45RB expression based on a previously established protocol (Ostanin et al., 2009). Histograms and frequencies for CD45RB^hi^ cells are shown in cyan, histograms and frequencies for CD45RB^lo^ cells are shown in red.

2. The effect of IL-2REH during the colitis induction by Teff of Teff + Treg.

We have now performed transfer colitis experiments in which CD45RB^hi^ cells were co-transferred with or without Tregs and mice were dosed with IL-2-REH at the onset of weight loss. For these experiments, we only administered IL-2-REH to mice which received Tregs as we have previously shown that IL-2-REH does not induce the differentiation of Tregs from Tcons (Figure 2A-C). Under these conditions, IL-2-REH expands Tregs but does not provide a therapeutic benefit (Figure 6—figure supplement 2B/C). In fact, IL-2-REH administration at this late time point exacerbated disease, likely by a small degree of potentiation of CD45RB^hi^ cells which become activated (CD25+) during colitis induction. We do not think the lack of therapeutic efficacy in this experiment compromises the message of the paper or the value of IL-2REH as a new class of mechanism-based Treg potentiators with possible utility in autoimmune diseases. In the clinical situation IL-2REH will be administered systemically, as we did for the DSS model.

More generally, we wish to emphasize that our collective findings in the paper were more to highlight a new class of cell-selective IL-2 analogs, and the concept of exploiting cell type intrinsic response thresholds by cytokines in vivo, which certainly has therapeutic relevance as we have shown, but the paper falls far short of definitive proof of IL-2REH utility as a drug for autoimmune disease. Indeed, the clinical utility of Treg expansion by any means for treating autoimmunity, remains to be determined. We could make minor text modification to the manuscript to better balance the conceptual advances with the clinical implications.

3. Another essential experiment is to test lower doses of IL-2 as indicated by reviewer 3.

The dose of IL-2 used in the DSS colitis experiments (5mg of MSA-hIL-2, every 2 days) is similar to that used in low-dose IL-2 regimes for the treatment of diabetes in NOD mice when adjusted for the molecular weight of albumin (Grinberg-Bleyer et al., 2010). However, we have also performed DSS experiments at even lower doses of IL-2 similar to those used in the treatment of systemic lupus erythematosus (SLE) (He et al., 2020). Administration of ultra-low-dose IL-2 does not provide a therapeutic benefit in these experiments (Author response image 2). We have updated the text to include explanation of dosing in the original DSS experiment (Page 14, lines 17-19).

**Author response image 2. sa2fig2:** Effects of IL-2 dose on DSS-induced colitis. C57BL/6J mice were placed on 3.5% DSS in drinking water for 6 days. Starting on day 4, mice received subcutaneous injections of PBS (n=10), 5mg IL-2 (n=10), 1mg IL-2 (n=10) or 0.2mg IL-2 (n=10) every other day for 3 doses. Data were compared to mice maintained on drinking water without DSS (n=5). Data are expressed as mean +/- SEM.

Reviewer #1:Bottom of page 6- the description of the single molecule tracking experiment to investigate receptor dimerisation is not clear. It seems that they have expressed β and common γ chains with the same GFP mutant, but them somehow tagged them with different dye labelled nano bodies and then tracked these on the surface to track movement of heterodimers. The authors need to better explain how β and common γ chains are selectively labelled- there must be some difference in the GFP tags or this would/could not work. The term co-locomotion is unconventional as it suggests that the receptors are self-propelled on the surface of the cell. It should be possible to determine if the motion is diffusive and then co-diffusion. If there is an active component it would more typically be referred to as transport. Locomotion is a term more reserved for movement of cells.

We thank the reviewer for pointing out this potential area of confusion surrounding the receptor tracking experiments. For specific labeling of IL-2Rb and g_c_, we have used two engineered variants of GFP, which are orthogonally recognized by two different anti-GFP nanobodies, enhancer and minimizer. In the revised manuscript, we have included a more detailed description of this approach in the method section and main text (Page 6, lines 18-21) as well as Figure 1C-D and the corresponding part of the figure legend. Further, as suggested by the reviewer we have changed the term “co-locomotion” to “co-diffusion”.

Bottom of page 9- The way the authors refer to recipient and donor seems is written as if it refers to the mice, rather than the recipient (endogenous CD25+ and CD25-) or donor (CD25-) cells. The authors just need to clarify that it’s the cells they are referring to. The logic of the experiment is otherwise sound.

We thank the reviewer for this note. The text has been updated to clarify this point (Page 10, line 6).

Bottom of page 13- they state that IL-2 REH mirrors two key features of IL-2 sensitivity in Treg. It’s not clear if "mirrors" is the correct term here. Would a better term be "exploits" in selective expansion of Treg in the steady state.

We agree with the reviewer and have changed to wording of this sentence (Page 14, line 2).

The authors use a DSS colitis model, which is not a classical autoimmune colitis, but more of a transient barrier disruption leading to inflammation that is influenced by T cells, but not directly T cell mediated as transfer colitis model. The in vivo results would be greatly clarified by repeating these points with the transfer colitis model.

As discussed in our comment to ‘essential revision #1’ above, the high frequency of Tregs among CD45RB^lo^ cells makes comparison of IL-2-REH expended Tregs difficult in the transfer colitis model. Given that we see similar suppression of Tcon proliferation by Tregs isolated from PBS, IL-2 and IL-2-REH treated mice and that Tregs already represent a high proportion of CD45RB^lo^ cells in untreated mice, we would not expect to see differences between groups using the T cell transfer colitis model with pre-expanded Tregs.

Reviewer #2:Many years ago, the laboratory of Chris Garcia embarked into a powerful research line under the general concept of "separating" the different functions of IL-2 from adverse effects as well as, more recently, the differential effects on regulatory T cells (Treg) and effector T cells and NK cells.For example, some years ago the authors described that IL-2 activity was greater on CD25HI cells (e.g. Treg) in the presence of JES6-1, and greater on CD25LO cells (e.g. CD8^+^ T cells) in the presence of S4B6. JES6-1 and S4B6 are two different anti-IL-2 antibodies.Chris Garcia's lab also was involved in the generation of several IL-2 muteins, in which the Il2 gene was mutated to improve IL-2 binding to CD122. This generated an IL-2 molecule that was relatively more capable of expanding CTL than Treg, a "superkine" that was, at the time, sought after by cancer immunologists.In their laboratory's current iteration, Dr. Garcia and co-workers added a number of mutations that weaken the binding to IL2Rg (CD132/Common gc). Their successful structural predictions led to IL-2 muteins that display reduced STAT5 phosphorylation at saturating mutein doses. This manuscript is another useful attempt to understand the differential effects of IL-2, and the practical implications.Of the shown IL-2 muteins, IL-2-REH is the one with highest phospho-STAT5-inducing activity. It is also the one that induces the highest expansion of Treg cells. Comparatively with IL-2, IL-2-REH is a better inducer of Treg expansion and a poor inducer of CTL expansion and gene expression. Other muteins may have too low an activity on IL-2 receptor to trigger any of the IL-2 functions. The dependence of IL-2-REH on IL2ra for signaling was shown by a number of independent ways, and is convincing.1. This is not an issue unique to this paper, but it is unclear why IL-2, but, to some degree also IL-2-REH, cause splenomegaly with granulocytosis. Granulocytes don't express IL-2 receptors, so why are they recruited so early and in such high numbers? There must be a very powerful proinflammatory signal produced by NK cells or conventional T cells in response to IL-2. If IL-2-REH is to become a therapy, it would require repeated administration, and, perhaps, it would induce granulocytosis as well in a chronic setting. The readers will benefit from a discussion on this topic.

We agree with the reviewer that the expansion of granulocytes upon systemic IL-2 administration is most likely a secondary effect induced by NK cell or T cell secretion of pro-inflammatory cytokines. This effect is greatly attenuated with IL-2-REH such that there is not a significant difference in granulocyte frequency or number following REH administration. We have updated the text to discuss this point (Page 8, lines 7-10).

2. Does IL-2-REH induce IL2ra expression in conventional CD4 cells and CD8 T cells, as does IL-2? The blasts used in Figure 5 are not generated using IL-2 muteins, and IL2ra did not make it to the top 30 genes in Figure 4, so I can't tell. If IL2ra expression is induced by IL-2-REH in conventional T cells, wouldn't one expect that, after a brief delay, IL-2-REH would lead to the proliferation of conventional T cells as well?

Expression of *Il2ra* is more strongly induced by IL-2 than IL-2-REH in CD8^+^ T cells following ex vivo stimulation (Author response image 3). Consistent with this effect, IL-2Ra surface expression on CD4^+^FoxP3- cells was elevated by IL-2 but not IL-2-REH (Figure 2—figure supplement 2 and response to point 4 below). Given that our in vivo experiments last from 3-7 days and we do not observe significant expansion of conventional cells with IL-2-REH, we think it unlikely that IL-2-REH induces sufficient expression of IL-2Ra to enhance sensitivity of conventional T cells.

**Author response image 3. sa2fig3:** Expression of *Il2ra* in CD8+ T cells following 4-hour stimulation with 200nM cytokine. Reads per kilobase per million (RPKM) were normalized to unstimulated CD8+ T cells.

3. It is important to characterize the duration of Treg increase (also quality) in vivo, much beyond day 7 post IL-2-REH treatment beginning.

We have performed tracking experiment to monitor the kinetics of Treg expansion and contraction. Following dosing of cytokine on day 0, 3 and 6, Tregs remain elevated for three days followed by contraction back to baseline levels 1 week after cessation of cytokine administration. This data has been added Figure 6—figure supplement 1C.

4. From the early Foxp3 days, it was apparent that there is a subset (10-20%) of Foxp3+ Tregs that are CD25-low, in a continuum pattern. In Dr. Garcia's data, both the IL-2 and IL-2-REH expanded Tregs express higher surface IL2ra than the untreated cells or H9-REH-treated cells. Unless the effect is short-lived, the level (MFI) of surface IL2ra expression attained in each case may be correlating with the functionality of the Tregs, and could be another important parameter to consider for suppression of autoimmune disease in vivo.

The reviewer is correct to point out that IL-2 and IL-2-REH elicit increased IL-2Ra expression on Tregs. Despite these differences, Tregs from cytokine treated mice were similar in their suppressive capacity to Tregs from untreated mice (Figure 6—figure supplement 1). Instead, the major effect of IL-2-REH appears to be expansion of Tregs rather than increased suppressive capacity on a per-cell basis.

5. I don't understand the result in Figure 5H, left panel. SOCS1 inhibits IL-2 signaling, but, in this experiment, overexpression of SOCS1 does not appear to have any effect on STAT5 phosphorylation when IL-2 was added. Do CTLL-2 cells express so much intrinsic SOCS1, that overexpression becomes irrelevant? Perhaps the reverse experiment (Socs1 knockdown) should have been attempted. How are the basal levels of SOCS1 in Tconv and Treg, and how does IL-2 and IL-2-REH affects SOCS1 expression?

The observation that SOCS1 overexpression does not inhibit IL-2 signaling in CTLL-2 cells may be due to the fact that CTLL-2 cells depend on IL-2 signaling for survival. Previously, CTLL-2s have been characterized to be low in SOCS1 expression through a transcriptional repression mechanism (Gregorieff et al., 2000). Our results suggest that the level of SOCS1 achieved by overexpression is sufficient to suppress IL-2-REH signaling but not IL-2 signaling.

These experiments were designed to test the role of SOCS1 in suppression of IL-2-REH signaling as Tregs have previously been shown to express low levels of SOCS1 through a micro-RNA dependent mechanism (Lu et al., 2009). To the authors second question about the effects of cytokine on SOCS1 expression, we see that IL-2 induces *Socs1* expression in both Tregs and CD8^+^ T cells following 4-hour stimulation with 200nM cytokine while REH induces *Socs1* selectively in Tregs although to a lesser extent than seen with REH (Figure 6—figure supplement 2B/C). Given the acute nature of the stimulations used in Figure 5H-I, we think that feedback is unlikely to explain these results, however, induction of SOCS1 by IL-2 may serve to inhibit STAT signaling in vivo where cytokines are administered over a longer time scale.

**Author response image 4. sa2fig4:** Expression of *Socs1* in Tregs and CD8^+^ T cells following 4-hour stimulation with 200nM cytokine. Reads per kilobase per million (RPKM) were normalized to unstimulated cells.

6. The DSS Colitis experiment is somewhat disappointing. While IL-2-REH drives a striking increase of Treg in mLN and of Foxp3+Gzmb+ cells the colonic tissue, the level of protection from disease is not striking. What other variables could be participating in the amelioration? Perhaps another colitis model, such as the adoptive transfer model (also called the Powrie model), might be beneficial, as it would even test the stability of the transferred Tregs in host mice that did not receive IL-2 or IL-2 muteins.

We thank the reviewer for their comments on the effects of IL-2-REH in the DSS-induced colitis model. While IL-2-REH does not prevent induction of weight loss, these experiments show that IL-2-REH is capable of expanding Tregs in the context of an inflammatory response and reducing disease severity as measured by histological score and body weight. As mentioned in our response to ‘essential revision #1’, we feel that the transfer colitis model, while powerful, is unlikely to show differences between untreated and cytokine treated groups given the high prevalence of Tregs among CD45RB^lo^ cells in naïve mice. For this reason, we think that the DSS-induced colitis model and in vitro suppression assays shown in Figure 6 and Figure 6 —figure supplement 1 better capture the effects of IL-2-REH.

My conclusion is that this is an important contribution, but some experiments that are easy to perform could add valuable information to the manuscript.

We thank the reviewer for their constructive suggestions.

Reviewer #3:First paragraph of the introduction lacks references.Page 3 Line 14: "severe"

This has been corrected.

Page 6 line 12: It is not clear what RETR is.

A description of H9-RETR has been added.

Figure 1 and S1: S1 pSTAT5 MFI shows that human IL-2 saturates the pSTAT5 at 100 nM while H9 (high affinity variant for IL-2Rb) saturates it at 102 nM. This seems to contradict with the justification of the use of H9 instead of WT IL-2. However, towards the end of the Results section H9 is more properly described and referenced as "IL-2 variant with high affinity for IL-2Rb and reduced dependence on IL-2Ra (Levin et al.,2012)". This should come upfront at Figure 1 for clarity. Also, if there are TIRF and SPR data for WT IL-2, they can replace the H9 data in Figure 1 or be added to Supplementary Figure 1 as IL-2Ra dependency of REH variants may account for some differences in those readouts too.

We have added a more complete description of super-2 (page 6, line 1). The use of super-2 for initial characterization was to account for differences in EC50 between IL-2 and partial agonists (compare Figure 1B and Figure 1—figure supplement 1). These differences are likely due to a high level of cooperativity in the binding of IL-2Rb and gc (Wang et al., 2005). The biochemical data looking at receptor dimerization was performed in the absence of IL-2Ra to focus on the impact of gc-interface mutations on IL-2 receptor dimerization. In these cases, super-2 is necessary to allow comparison of partial agonists.

Figure 2 and S2: In Figure 2 A-D and S2 B-F, PBS injected spleen percentages add up to more than 100% while the total numbers don't match the T and B percentages. The graphs may come from different experiments, so authors should clarify the mismatches and replace the graphs.

Thank you for pointing out this potential area of confusion. In our submission, the frequency of B, NK and granulocytes were expressed as a percentage of CD3- cells while the frequency of CD3+ cells is expressed as a % of CD45+. We have updated the graphs to express all populations as a percent of CD45+ for simplicity.

Figure 2 would also benefit from showing a decrease in the total numbers of Foxp3-CD4^+^ effector cells in complementary to the increase in Treg #. IFNγ production of CD4^+^ T cells can also be added to the supplementary figures.

Data showing the frequency and number of FoxP3^-^IL-2Ra^+^ cells has been added to Figure 2 —figure supplement 2C-E. As would be predicted, we see elevated IL-2Ra+ Tcons following treatment with IL-2 but not IL-2-REH. For intracellular cytokine stain in CD4^+^ T cells, we did not observe an IL-2 dependent increase in IFNg+ cells following PMA-Ionomycin treatment, indicating that the requirements for IFNg production in CD4 and CD8^+^ T cells are likely different.

P8 Line 14: "increased cytotoxicity" instead of "increase cytotoxicity"

This has been corrected.

Figure 3-4-5: Well-performed experiments with clear findings.

We thank the reviewer for their positive comments.

Figure 6:The data presented by the authors are promising, albeit brief. In the colitis model, although IL-2-REH treated mice displayed faster and greater recovery, especially at later time points, findings may be reflective of the IL-2 dose administered. DSS colitis is a chemical injury model that represents the physiology of mucosal healing in human ulcerative colitis, however it is inferior to CD45RBhi CD4^+^ T cell transfer colitis model in the way it represents the contribution of effector and regulatory T cells.

We than the reviewer for their comments about the dose of IL-2 used in these experiments. As explained in our response to ‘essential revision #3’, the 5mg dose of MSA-IL-2 used in these experiments is similar to that used previously to assess low-dose IL-2 in treatment of non-obese diabetic mice when adjusted for the molecular weight of albumin (Grinberg-Bleyer et al., 2010). We have further characterized the impact of ultra-low dose IL-2 on DSS-induced colitis, however, we do not find enhanced benefit at 1mg or 0.2mg of MSA-IL-2 (Author response image 2). The text has been updated to include an explanation of the dosing strategy used in the DSS experiments (page 14, lines 17-19).

While we agree with the reviewer that the role of Tregs in the T cell transfer colitis model has been well established, the high frequency of Tregs among CD45RB^lo^ cells in naïve mice means that IL-2-REH is unlikely to provide additional benefit relative to Tregs from PBS treated mice. For this reason, we think the DSS-induced colitis model in conjunction with the in vitro suppression presented in Figure 6 —figure supplement 1 better highlight the benefit of IL-2-REH expanded Tregs.

In order to reach a fair conclusion about the therapeutic effect of IL2-REH on colitis, it should also be compared to lower doses of IL-2. The in vitro findings provided in Figure 5 supplement 1 suggest that the negative effects of IL-2 can be alleviated or at least matched to that of IL2-REH at low IL-2 doses. Figure 5 supplement 1 indicates that minimum IL-2 dose (100 nM) that provides maximum pSTAT5 signaling in Tregs doesn't induce signaling in any of the effector cells except for the CD4 blasts. Therefore, a more ideal approach for characterizing the therapeutic effects of IL2-REH in vivo should involve comparison of different IL-2 and IL-2-REH doses in CD45RBhi CD4^+^ T cell transfer colitis model. In such a system, the activity of CD45RBlow CD4^+^ regulatory cells can be tested with or without IL-10 blockade, however blockade experiment is not crucial.

As mentioned above, the dose of IL-2 used in the DSS-colitis experiments (5mg of MSA-IL-2 every other day) is similar to the previously established ‘low dose IL-2’ used for the reversal of non-obese diabetes when adjusted for the molecular weight of albumin fusion (Grinberg-Bleyer et al., 2010) and the text has been updated to include this information (page 14, lines 17-19). In addition, further reducing the dose of IL-2 similar to that used in the clinical treatment of SLE did not improve colitis outcome (Author response image 2).

While we agree that the T cell transfer colitis model is a powerful method for assessing the suppressive capacity of Tregs, as discussed in our response to ‘essential revision 1’, the high frequency of Tregs among CD45RB^lo^ cells will likely mask any effect of IL-2-REH expansion. For this reason, we think the DSS-induced colitis model is preferable.

[Editors' note: further revisions were suggested prior to acceptance, as described below.]

You have used structure based design to create IL-2-REH, a partial agonist that can promote preferential expansion of Treg in vivo. Testing IL-2-REH in two models of auto inflammatory disease- the DSS and transfer colitis models- you have found protection in the DSS model, but not the transfer colitis model, despite increase Treg numbers. The results are important to the field in providing a new approach to cytokine engineering and also pointing to limitations of Treg expansion in amelioration of auto inflammatory disease.The reviewers appreciate your effort to look at the transfer colitis model. The point that IL2REH can promote proliferation, yet not improve the clinical outcome in the model is an important point for the field and should be included in the final paper.1. Incorporate the transfer colitis data on Treg expansion and clinical scores in a supplemental figure and discuss the implications for the advance in selective Treg expansion, but remaining challenges to translate this into clinical impact.

Thank you for the summary of our work and suggestions about how to improve the manuscript. We agree that showing and discussing the apparent disconnect between Treg expansion and disease suppression in transfer colitis would be beneficial for readers. We now present that data in Figure 6 —figure supplement 2 and include a discussion of our findings (page 14 line 16 – page 15 line 11 and page 16 line 19-22).

Reviewer #1:The authors have evaluated IL-2-REH in a different colitis model and tested a range of doses in the DSS model. The results support the ability of IL-2-REH to selectively expand Treg, but the clinical scores in the transfer colitis model were worse with IL-2-REH, even as Treg expansion was increased. The authors have addressed a number of minor issues in the text and added additional discussion that improve the manuscript. The authors have addressed a challenge related to selective expansion of Treg in vivo, but the limitation is that expansion of Treg may not be sufficient, in all cases, to achieve remission of inflammatory disease.

We thank the reviewer for their thoughtful evaluation of our work. We have updated the text to include a more complete explanation of our finding that Treg expansion does not necessarily correlate with disease outcome in the T cell transfer colitis model (page 14 line 16 – page 15 line 11).

Reviewer #3:While the authors have spent praiseworthy effort on revising the key points raised by reviewers, some points require further clarification and revision. Because of the marginal effects of both IL-2 and IL-2-REH in DSS colitis and slightly disease-exacerbating potential of IL2-REH in the adoptive transfer colitis model, one should be extra cautious to mention the therapeutic effects of this particular IL-2 mutein in colitis. One of the reasons for the unexpected outcome in transfer colitis might actually be due to the timing of initial cytokine dose (day 24) which could have tipped the balance off towards effector activation. Therefore, it would be useful to share transfer colitis data with readers adding also a relevant statement into Discussion section stating the reservations about therapeutic effects of IL-2-REH, at least in the colitis models tested.

We agree that a more complete discussion of the in vivo effects of IL-2-REH specifically and Treg expansion more generally would be of use. To this end, we now present the transfer colitis data in Figure 6 —figure supplement 2 and discussion of differences in DSS and transfer colitis models (page 16 lines 19-22).

I agree that the transfer colitis model, while powerful, is unlikely to show differences between untreated and cytokine treated groups given the data presented in Figure 6—figure supplement 2. However, instead of replacing it with DSS-induced colitis model that only shows some marginal effect and in vitro suppression assays one should rather consider titrating the transferred CD45RBlo/Foxp3.GFP numbers down in transfer colitis setting. This may provide some room for Treg expansion and the effects of cytokine treatment may be visible if a regimen with earlier cytokine doses is adapted. Furthermore, the in vitro suppression assay graph in Figure 6—figure supplement 1 is insufficient without a "No Treg" control to evaluate the degree of suppression in the Treg+ groups.

We agree with the reviewer that under conditions of limiting Tregs, IL-2-REH may be able to provide Treg expansion such that colitis is suppressed. Indeed, this was the aim of the transfer colitis in Figure 6—figure supplement 2. We used a 1:30 CD45RB^hi^ to Treg ratio, however, Tregs were still suppressive even at this low ratio. While we agree that optimization of the transfer colitis model may demonstrate the efficacy of IL-2-REH treatment under conditions of limiting Tregs or with altered timing, we feel that the scope of these experiments is beyond that of the current study.

To the reviewers second point, a “No Treg” control was included in Figure 6 —figure supplement 1 (see upper FACS plots) but for clarity we have now included a dashed line in the plot to indicate the CellTrace Violet dilution in the absence of Tregs.